# ANALYSIS OF TASK TRANSFERABILITY IN LARGE PRE-TRAINED CLASSIFIERS

## ABSTRACT

Transfer learning is a cornerstone of modern machine learning, enabling models to transfer the knowledge acquired from a source task to downstream target tasks with minimal fine-tuning. However, the relationship between the source task performance and the downstream target task performance (i.e., transferability) is poorly understood. In this work, we rigorously analyze the transferability of large pre-trained models on downstream classification tasks after linear fine-tuning. We use a novel Task Transfer Analysis approach that transforms the distribution (and classifier) of the source task to produce a new distribution (and classifier) similar to that of the target task. Using this, we propose an upper bound on transferability composed of the Wasserstein distance between the transformed source and the target distributions, the conditional entropy between the label distributions of the two tasks, and the weighted loss of the source classifier on the source task. We propose an optimization problem that minimizes the proposed bound to estimate transferability. Using state-of-the-art pre-trained models, we show that the proposed upper bound accurately estimates transferability on various datasets and demonstrate the importance of relatedness between the source and target tasks for achieving high transferability.

## 1 INTRODUCTION

Transfer learning (Pan & Yang, 2010; Weiss et al., 2016) is a powerful tool for developing high-performance machine learning models in scenarios when sufficient labeled data is unavailable or when training large models is computationally challenging. Since large pre-trained models are becoming the cornerstone of machine learning (Radford et al., 2021; Chen et al., 2020; 2021; Devlin et al., 2019), understanding how these models improve the performance of a downstream task is crucial. While the large pre-trained models achieve high performance even in the zero-shot inference setting (Radford et al., 2021), their performance can be improved by fine-tuning them on downstream tasks. However, due to the size of these models, fine-tuning all the layers of the models is computationally challenging and expensive. On the other hand, fine-tuning/learning a linear layer on top of the representations from these models is both efficient and effective and is thus, the focus of our work.

While transfer learning has achieved remarkable success the relationship between the performance of the model on the source and the target task is not well understood. Moreover, previous analytical works such as those based on domain adaptation (Ben-David et al., 2007; 2010; Shen et al., 2018; Mansour et al., 2009; Le et al., 2021; Mehra et al., 2021; 2022) are not applicable to transfer learning since both the feature and label sets can change between the source and target tasks in this setting. Recently, Tran et al. (2019) showed that when only label sets change between the source and target tasks, the performance gap between the two tasks can be bounded by conditional entropy between the label sets. However, their analysis is in a limited setting and is not applicable to a general cross-domain (different features/priors) cross-task (different label sets) setting. Another recent line of work, score-based transferability estimation (SbTE) (You et al., 2021; Tan et al., 2021; Huang et al., 2022; Nguyen et al., 2020) focuses on developing scores that can be computed more efficiently than fine-tuning and are correlated with transferability. While these scores are useful for selecting the pretrained model that can produce the best transfer of performance on a specific downstream task, they give limited insights into how transferability relates to the performance of the source task.

Thus, we propose a novel Task Transfer Analysis approach to analyze transferability in a general setting in relation to the performance of the source task. Our approach works by transforming the source distribution (and the classifier of the source task) by transforming the class-prior distribution, label set, and feature space to obtain a new distribution (and classifier) that is similar to that of the target task (Fig. 1). Under this transformation model, we first show that the performance of the transformed source relates directly to the performance of the original source task (Theorem 1). Next, since the distribution of the target task may not be a transformation of the source task, there could still be a performance difference between the transformed source and the target distribution. We use the Wasserstein distance (Peyré et al., 2019; Villani, 2009) between the transformed source and target distributions to explain the residual performance difference. This leads to our Theorem 2, which relates transferability to the performance of the transformed source distribution and type-1 Wasserstein distance between the distributions of the target and the transformed source. Using these we upper bound transferability as a sum of three terms (Theorem 3), namely, re-weighted source loss (due to a class prior distribution differences between tasks), label mismatch (as conditional entropy between the label distributions of the tasks) and distribution mismatch (as Wasserstein distance).

To effectively approximate the upper bound on transferability, we propose an optimization problem to learn the transformations to transform the source task (and classifier) to be similar to that of the target task. The optimization works by minimizing the proposed upper bound. Using state-of-the-art (SOTA) pre-trained models with different architectures and trained with various training methods on computer vision and natural language processing (NLP) tasks, we show that our upper bound closely predicts actual transferability (Sec. 4.2 and 4.4). Unlike our bound, scores from SbTE methods are not comparable across tasks. Moreover, SbTE scores are relative measures that are meaningful only when compared with the scores of different pretrained models on the same task (e.g., a score of 1 for CIFAR100 does not indicate whether transferability is good or bad), whereas our bound is an absolute measure (e.g., an upper bound of 1 on CIFAR100 implies that cross-entropy loss will be less than 1, indicating good transferability). Furthermore, when the source and target tasks are related (as measured by a small Wasserstein distance), their performance is provably similar. This enables us to accurately estimate the transferability to unseen target tasks that are related to the source task.

Our main contributions are summarized as follows:

- We rigorously analyze transferability for classification tasks. To the best of our knowledge, we propose the first upper bound on transferability in terms of the performance of the source task, in a cross-domain cross-task setting.

- We propose and solve a novel optimization for task transfer analysis to estimate transferability. By relating transferability to the performance of the source task we can estimate transferability based on the distance of the transformed source to unseen tasks.

- We demonstrate that the prediction from our bound is close to the actual transferability of SOTA classifiers trained with different architectures and pre-training methods on various datasets.

## 2 RELATED WORK

**Transfer learning:** Transfer learning achieves promising results across many areas of machine learning (Pan & Yang, 2010; Weiss et al., 2016) including NLP (Devlin et al., 2019; Sanh et al., 2020) and computer vision (Ren et al., 2015; Dai et al., 2016). Recent works have demonstrated that the transferability of models improves when trained with pretraining methods such as adversarial training (Salman et al., 2020), self-supervised learning (Chen et al., 2020; Caron et al., 2020; Chen et al., 2021) and by combining language and image information (Radford et al., 2021). The success of these is explained in terms of different training methods helping learn an improved feature representation. However, unlike our work, a rigorous analysis of transferability is not addressed in these works.

**Analytical works for learning under distribution shifts:** Prior works (Ben-David et al., 2007; 2010; Shen et al., 2018; Mansour et al., 2009; Le et al., 2021; Mehra et al., 2021; 2022) analytically explained learning under distribution shifts in terms of the distributional divergence between the marginal distributions and a label mismatch term. However, these results are applicable under assumptions such as covariate shift or label shift which need not be satisfied by transfer learning where both the data distribution and the label spaces can be different (see App B for detailed comparison). Other works (Ben-David & Schuller, 2003; Ruder, 2017; Padmakumar et al., 2022)

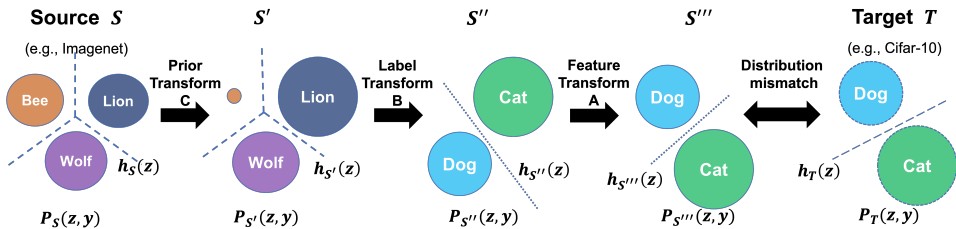

Figure 1: **: Overview of our Task Transfer Analysis approach:** A series of transformations are applied to the source distribution $P_S(z, y)$ and classifier $h_S$ to produce the transformed distribution $P_{S'''}$ and classifier $h_{S'''}$ making it comparable to those of a downstream target task. Prior transform $(S \to S')$ changes the class prior of the source distribution (e.g, an irrelevant Bee class in $S$ now has smaller prior) followed by label set transform $(S' \to S'')$ (e.g., to match {Lion, Wolf} with {Cat, Dog}), followed by feature space transform $(S'' \to S''')$ to match the feature distribution $P_T(z, y)$.

showed the success of learning in the multi-task setting using task-relatedness. These works showed that when tasks are weakly related, learning a single model does not perform well for both tasks. Our analytical results illustrate the same effect of task-relatedness on transferability (Sec. 4.3). Recently, Tran et al. (2019) proposed an upper bound on transferability in a restrictive setting of the same features for the source and target task. Our analysis does not make any such assumption.

**Score-based transferability estimation (SbTE):** These works (Bao et al., 2019; Nguyen et al., 2020; Huang et al., 2022; You et al., 2021; Tan et al., 2021) use data from the target task to produce a score correlated with transferability. This score is designed for choosing the best pre-trained model from a model zoo that should be used for transfer learning to a specific target task without actual fine-tuning. Tran et al. (2019) proposed the Negative Conditional Entropy (NCE) score that predicts transferability using the negative conditional entropy between labels of the tasks. Nguyen et al. (2020) proposed the LEEP score and computed NCE using soft labels for the target task from a pre-trained model. OT-CE (Tan et al., 2021) combined Wasserstein distance (Alvarez-Melis & Fusi, 2020) and NCE whereas (Bao et al., 2019; You et al., 2021) estimate likelihood and the marginalized likelihood of labeled target examples to estimate transferability. While these scores are useful for a practical problem, the goal of these works is not to derive a tight upper bound on transferability, study the relation between transferability and the performance on the source task, or understand how relatedness between the source and target tasks affects transferability (see App. B for detailed comparison).

## 3 ANALYSIS OF TASK TRANSFERABILITY

**Problem setting and notations:** Let $P_S(x, y)$ and $P_T(x, y)$ denote the distributions of the source and the target tasks, defined on $\mathcal{X}_S \times \mathcal{Y}_S$ and $\mathcal{X}_T \times \mathcal{Y}_T$ respectively. We assume that the feature spaces are common ($\mathcal{X}_S = \mathcal{X}_T = \mathcal{X}$) such as RGB images, but the source label set $\mathcal{Y}_S = \{1, 2, \cdots, K_S\}$ and the target label set $\mathcal{Y}_T = \{1, 2, \cdots, K_T\}$ can be entirely different to allow for arbitrary (unknown) downstream tasks. We assume the number of source classes ($K_S$) is greater than or equal to the number of target classes ($K_T$). In the transfer learning setting, an encoder $g : \mathcal{X} \to \mathcal{Z}$ is first (pre)trained with source data with or without the source labels depending on the training method (e.g., supervised vs. self-supervised). We denote the resultant push-forward distributions of $S$ and $T$ on the encoder output space as $P_S(z, y)$ and $P_T(z, y)$. With a fixed representation $g$, a linear softmax classifier $h(z) : \mathcal{Z} \to \Delta$ that outputs a probability vector is learned for the source ($h_S$) and for the target ($h_T$) separately, where $\Delta_{S/T}$ is a $K_S/K_T$ simplex for $S/T$, respectively. The classifiers $h_S$ and $h_T$ are obtained by $\min_{h_S \in \mathcal{H}} \mathbb{E}_{(z,y) \in P_S}[\ell(h_S(z; g), y)]$ and $\min_{h_T \in \mathcal{H}} \mathbb{E}_{(z,y) \in P_T}[\ell(h_T(z; g), y)]$ where $\mathcal{H}$ is the set of linear classifiers (parametrized by $w$) and $\ell(h(z), y) = -\log(h(z)_y) = -w_y^T z + \log \sum_i e^{w_i^T z}$ is the cross-entropy loss. Proofs for Sec. 3 are in App. A.

### 3.1 THE TASK TRANSFER ANALYSIS MODEL FOR ANALYZING TRANSFERABILITY

The source and the target tasks share the same encoder but do not share label sets or data distributions. Therefore, to relate the two tasks, we propose a chain of three simple transformations: 1) prior

transformation (from $S$ to $S'$), 2) label transformation (from $S'$ to $S''$), and 3) feature transformation (from $S''$ to $S'''$). The $S', S'', S'''$ are intermediate domain names after each of the transformations are applied. The corresponding classifier in each domain is denoted by $h_{S'}$, $h_{S''}$, and $h_{S'''}$. This is illustrated in Fig. 1. The distribution after the transform $P_{S'''}$ has the same feature $\mathcal{Z}_{S'''} = \mathcal{Z}_T = \mathcal{Z}$ and label sets $\mathcal{Y}_{S'''} = \mathcal{Y}_T$ as the target task $T$, and consequently, the loss of the transformed classifier $h_{S'''}$ can be directly related to the loss of the target classifier $h_T$.

### 3.1.1 PRIOR TRANSFORMATION ($S \to S'$)

In transfer learning, it is common that the source task has more classes than the target task ($|\mathcal{Y}_S| \geq |\mathcal{Y}_T|$) and it is highly likely that many of the source classes are irrelevant for transfer to the target classes. For e.g., while transferring from Imagenet to CIFAR10, only a small portion of the source classes are relevant to the target classes. The prior transform accounts for the relative importance of the source classes. This is illustrated in Fig. 1 where changing the class prior of $S$ reduces the prior of the Bee class and increases the priors of Wolf and Lion classes (shown by the changed size of classes Wolf and Lion in $S'$). While transforming the prior of $S$, we keep the conditional distribution and the classifier the same i.e., $P_{S'}(z|y) = P_S(z|y)$ and $h_{S'}(z) = h_S(z)$. Lemma 1 shows that the expected loss of the classifier $h_S$ in $S'$ is a re-weighted version of the loss on the source domain $S$.

**Lemma 1.** *Let $C := \left[ \frac{P_{S'}(y)}{P_S(y)} \right]_{y=1}^{|\mathcal{Y}_S|}$ be a vector of probability ratios and the classifier $h_{S'}(z) := h_S(z)$, then we have $\mathbb{E}_{P_{S'}(z,y)}[\ell(h_{S'}(z), y)] = \mathbb{E}_{P_S(z,y)}[C(y)\ell(h_S(z), y)]$, for any loss function $\ell$.*

### 3.1.2 LABEL TRANSFORMATION ($S' \to S''$)

Next, we apply a label transform so that the label set of the new domain $S''$ matches the label set of the target domain. To change the label set of the domain $S'$, we specify the conditional distribution $B_{ij} := P(y_{S''} = i|y_{S'} = j)$ ($B_{ij} \in [0, 1], \forall i, j, \sum_i B_{ij} = 1, \forall j$). The label $y_{S''}$ of an example from the domain $S''$ can be obtained via $BP(y_{S'})$. This generative process doesn't require the feature, i.e., $P_{S''}(y_{S''}|y_{S'}, z) = P_{S''}(y_{S''}|y_{S'})$. $B$ with sparse entries (i.e., only one entry of a column is 1) models a deterministic map from $\mathcal{Y}_S$ to $\mathcal{Y}_T$; $B$ with dense entries models a weaker association. This process is illustrated in Fig. 1 which shows the map from {Bee, Wolf, Lion} $\subset \mathcal{Y}_{S'}$ to {Dog, Cat} $\subset \mathcal{Y}_T$ after applying the transform. Under this generative model, a reasonable choice of classifier for the new domain $S''$ is $h_{S''}(z) = Bh_{S'}(z)$. (Note that $h$ outputs a probability vector.) We show the conditions under which this classifier is optimal in Corollary 2 in App. A. Lemma 2 shows that the expected loss in the new domain $S''$ depends on the loss of the domain $S'$ and the conditional entropy between the label sets of the tasks $S'$ and $S''$.

**Lemma 2.** *Let $B$ be a $|\mathcal{Y}_S| \times |\mathcal{Y}_T|$ matrix with $B_{ij} = P(y_{S''} = i|y_{S'} = j)$ and $h_{S''}(z) := Bh_{S'}(z)$ and $\ell$ be the cross-entropy loss. Then, $\mathbb{E}_{P_{S''}(z,y)}[\ell(h_{S''}(z), y)] \leq \mathbb{E}_{P_{S'}(z,y)}[\ell(h_{S'}(z), y)] + H(\mathcal{Y}_{S''}|\mathcal{Y}_{S'})$, where $H(\mathcal{Y}_{S''}|\mathcal{Y}_{S'}) := [-\sum_{y_{S'} \in \mathcal{Y}_{S'}} \sum_{y_{S''} \in \mathcal{Y}_{S''}} P_{S'}(y_{S'})B_{y_{S''},y_{S'}} \log(B_{y_{S''},y_{S'}})]$ denotes the conditional entropy.*

### 3.1.3 FEATURE TRANSFORMATION ($S'' \to S'''$)

The final step involves changing the feature space of the distribution $S''$. We use an invertible linear transform via $A$ of the distribution in $S''$ to obtain the new distribution $S'''$. After the transform, the classifier associated with the new domain $S'''$ is $h_{S'''}(z) = h_{S''}(A^{-1}(z))$. This is illustrated in Fig. 1 after feature transform using $A$. Lemma 3 shows that a linear transform of the space and classifier does not incur any additional loss. In Corollary 3 in App. A we show the optimality of $h_{S''}$ implies optimality of $h_{S'''}$.

**Lemma 3.** *Let $A : \mathcal{Z} \to \mathcal{Z}$ be an invertible linear map of features and the classifier $h_{S'''}(z_{S'''}) := h_{S''}(A^{-1}(z_{S'''}))$. Then $\mathbb{E}_{P_{S'''}(z,y)}[\ell(h_{S'''}(z), y)] = \mathbb{E}_{P_{S''}(z,y)}[\ell(h_{S''}(z), y)]$ for any loss $\ell$.*

### 3.1.4 THREE TRANSFORMATIONS COMBINED

Combining Lemmas 1, 2, and 3 corresponding to the three transformations, we have the following.

**Theorem 1.** *Under the assumptions of Lemmas 1, 2, and 3 we have*

$$\mathbb{E}_{P_{S'''}(z,y)}[\ell(h_{S'''}(z),y)] \leq \underbrace{\mathbb{E}_{P_S(z,y)}[C(y)\ell(h_S(z),y)]}_{\textit{Re-weighted source loss}} + \underbrace{H(\mathcal{Y}_{S''}|\mathcal{Y}_{S'})}_{\textit{Label mismatch}}.$$

Theorem 1 provides an upper bound on the loss of the final transformed classifier/distribution in terms of the loss of the source classifier/distribution. The *re-weighted source loss* shows that the performance of the transformed classifier on the new domain is linked to the label-wise re-weighted loss of the source classifier on the source domain. This implies that one can use only the relevant source classes to contribute to the transferability bound. The second term *label mismatch* shows that the performance of the distribution $S'''$ and $S$ depends on the conditional entropy $H(\mathcal{Y}_{S''}|\mathcal{Y}_{S'}; B)$ between the label distributions of the domain $S''$ and $S'$. A high value of $H$ implies that the labels of the source do not provide much information about the labels of the target leading to lower transferability, whereas a low $H$ implies higher transferability. Corollary 1 below shows the case when the bound in Theorem 1 becomes equality. In particular, when the number of classes is the same between $S$ and $S'''$ and there is a deterministic mapping of the classes of the two domains, conditional entropy can be minimized to zero making the bound equality.

**Corollary 1.** *Let $e$ be one-hot encoding of the labels, $|Y_{S'''}| = |Y_S|$, $B : \Delta_{S'} \to \Delta_{S''}$ be a permutation matrix and $y_{S''} := \sigma(y_{S'}) := \arg\max_{y \in \mathcal{Y}_{S''}}(B(e(y_{S'})))_y$ then under the assumptions of Lemmas 1, 2, and 3 we have $\mathbb{E}_{P_{S'''}(z,y)}[\ell(h_{S'''}(z),y)] = \mathbb{E}_{P_S(z,y)}[C(y)\ell(h_S(z),y)]$.*

## 3.2 DISTRIBUTION MISMATCH BETWEEN $P_{S'''}$ AND $P_T$

After the three transformations, the transformed source $P_{S'''}(z,y)$ can now be compared with the target $P_T(z,y)$. However, these are only simple transformations and $P_{S'''}$ cannot be made identical to $P_T$ in general. This mismatch between $P_{S'''}$ and $P_T$ can be measured by the Wasserstein or Optimal Transport distance (Peyré et al., 2019; Villani, 2009). Many prior works have provided analytical results using Wasserstein distance for the problem of learning under distribution shift (see App. B) Since our goal is to match two joint distributions defined on $\mathcal{Z} \times \mathcal{Y}$ we use

$$d((z,y),(z',y')) := \|z - z'\|_2 + \infty \cdot 1_{y \neq y'}, \text{ where } z, z' \in \mathcal{Z} \text{ and } y, y' \in \mathcal{Y} \tag{1}$$

as our base distance (Sinha et al., 2017) to define the (type-1) Wasserstein distance

$$W_d(P,Q) := \inf_{\pi \in \Pi(P,Q)} \mathbb{E}_{((z,y),(z',y')) \sim \pi}[d((z,y),(z',y'))]. \tag{2}$$

With this base distance, the Wasserstein distance between the joint distributions is the weighted sum of the Wasserstein distance between conditional distributions ($P(z|y)$) (Lemma 4 in App. A). Theorem 2 explains the final gap between the loss incurred on the transformed distribution $S'''$ and the target distribution $T$ due to the distribution mismatch.

**Assumption 1.** *1) The composition of the loss function and the classifier $\ell \circ h$ is a $\tau-$Lipschitz function w.r.t to $\|\cdot\|_2$ norm, i.e., $|\ell(h(z),y) - \ell(h(z'),y)| \leq \tau\|z - z'\|_2$ for all $y \in \mathcal{Y}$, $z, z' \in \mathcal{Z}$ where $h \in \mathcal{H}$. 2) The two priors are equal $P_T(y) = P_{S'''}(y)$.*

The assumption 2), can be satisfied since we have full control on the prior $P_{S'''}(y)$ via $B$ and $C$.

**Theorem 2.** *Let the distributions $T$ and $S'''$ be defined on the same domain $\mathcal{Z} \times \mathcal{Y}$ and assumption 1 holds, then $\mathbb{E}_{P_T(z,y)}[\ell(h(z),y)] - \mathbb{E}_{P_{S'''}(z,y)}[\ell(h(z),y)] \leq \underbrace{\tau\, W_d(P_{S'''}, P_T)}_{\textit{Distribution mismatch}}$, with $d$ as in Eq. 1.*

Theorem 2 shows that $\ell \circ h$ is $\tau-$Lipschitz then the performance gap between the transformed source distribution and the target distribution is bounded by the type-1 Wasserstein distance between the two distributions. This result is in line with prior works that provided analytical results on performance transfer in the domain adaptation literature. The Lipschitz coefficient of the composition can be bounded by $\tau$, by penalizing the gradient norm w.r.t $z$ at training time. Thus, for linear fine-tuning, we train the classifiers $h_S$ and $h_T$ with an additional gradient norm penalty $\max\{0, \|\nabla_z \ell(h(z),y)\|_2 - \tau\}$ to make them conform to the Lipschitz assumption (see App. C.5). Note that constraining the Lipschitz constant restricts the hypothesis class. The trade-off between the Lipschitz constant and the performance of $h$ is empirically evaluated in App. C.5.1.

### 3.3 FINAL BOUND

Here, we combine the results obtained in Theorem 1 and Theorem 2. The final bound proposed in Theorem 3 is one of our main contributions which explains transferability as a sum of three interpretable gaps which can be numerically approximated as will be shown in Sec. 4.

**Theorem 3.** *Let $\ell$ be the cross entropy loss then under the assumptions of Theorems 1 and 2 we have,*

$$\mathbb{E}_{P_T(z,y)}[\ell(h_T(z),y)] \leq \underbrace{\mathbb{E}_{P_S(z,y)}[C(y)\ell(h_S(z),y)]}_{\text{Re-weighted source loss}} + \underbrace{H(\mathcal{Y}_{S''}|\mathcal{Y}_{S'})}_{\text{Label mismatch}} + \underbrace{\tau\,W_d(P_{S'''},P_T)}_{\text{Distribution mismatch}}.$$

The theorem shows that transferability can be decomposed into the loss incurred while transforming the class prior distribution, label space, and feature space of the source distribution (first two terms) and the residual distance between the distribution generated through these transformations and the actual target distribution (last term). When the distribution of the target task is a transform of the source task then there exist transformations $A, B$, and $C$ such that the distribution $S'''$ matches the distribution of the target task exactly and the $W_d(P_{S'''}, P_T) = 0$. When labels are deterministically related (Corollary 1) the bound becomes an equality. Finally, while we analyze linear fine-tuning for its simplicity, our bounds hold for non-linear classifiers and non-linear feature transformations as well (see App. A.4).

## 4 EXPERIMENTS

In this section, we present an empirical study to demonstrate the effectiveness of the proposed task transfer analysis in predicting transferability to downstream tasks. We start by describing the optimization problem that allows us to transform the source distribution (and its classifier) to explain transferability. Next, we demonstrate the effectiveness of learning the transformation parameters by solving the proposed optimization problem on simple transfer tasks. Following this we demonstrate how the relatedness between the source and target tasks impacts transferability. Finally, we present a large-scale study showing a small gap between predicted and empirical transferability on classifiers with different architectures and trained using various training algorithms on vision and NLP tasks. Due to space limitation, NLP results are in the App. C, and experimental/dataset details in App. D.

### 4.1 OPTIMIZATION PROBLEM TO LEARN THE TRANSFORMATIONS

Here we describe the optimization problem (Eq. 3) to learn the transformations $A, B$, and $C$ to convert the source to the target task. We use two new variables: the inverse of the transformation $A$, denoted by $\bar{A} := A^{-1}$ and a new source prior distribution denoted by $D(y) := C(y)P_S(y)$. We solve the Eq. 3 with samples from the training data of the source and target tasks. Using test data from the source task and the learned transformations we measure the upper bound on transferability.

$$\min_{A,\bar{A},B,D} \quad \mathbb{E}_{P_S(z,y)}\left[\frac{D(y)}{P_S(y)}\ell(h_S(z),y)\right] + H(\mathcal{Y}_{S''}|\mathcal{Y}_{S'};B,D) + \tau W_d(P_{S'''},P_T;A,B) \text{ such that}$$

$$A\bar{A} = \bar{A}A = I, \ P_T(y) = BD, \ \sum_i B_{ij} = 1 \ \forall j, \ \sum_{y\in\mathcal{Y}_S} D(y) = 1, \ B_{ij} \in [0,1] \ \forall i,j, \ D_i \in [0,1] \ \forall i.$$

$$(3)$$

Our algorithm for task transfer analysis shows how we solve this optimization problem. The first three terms in Step 4 of our algorithm correspond to the terms in the objective of Eq. 3 while the two additional terms are added to penalize the constraints of class prior matching $P_T(y) = BD$ and invertibility of the matrix $A$, respectively as required by Theorem 3. We use the softmax operation to ensure $B$ and $D$ are a valid probability matrix and vector. In Step 3 of the algorithm, we use the network simplex flow algorithm from POT (Flamary et al., 2021) to compute the optimal coupling for matching the distributions $S'''$ and $T$. Since computing the Wasserstein distance over the entire dataset can be slow, we follow (Damodaran et al., 2018) and compute the coupling over batches. Using the optimal coupling from Step 3, we use batch SGD in Step 4 of the algorithm. Note that we use a base distance defined in Eq. 1 in Step 3, which is non-differentiable. Therefore in Step 4, we use a differentiable approximation $\tilde{d}((z,y),(z',y')) := d_{features}(z,z') + \nu \cdot \|Be(y) - e(y')\|_2$

(with $\nu = 10^8$) where $(z, y)$ and $(z', y')$ are samples from the domains $S$ and $T$ and $e(\cdot)$ denotes the one-hot embedding of the labels. We show in Fig. 7 in App. C.1.3 how the upper bound is minimized as the optimization learns the parameters of the transformations. When the target task has fewer classes, the algorithm adjusts the prior $D$ to select a similar number of source classes leading to a reduction in the re-weighted source loss. On one hand, the conditional entropy contributes to a sparse selection of the source classes, while on the other, the condition $P_T(y) - BD$ and the Wasserstein distance in Step 4 prevents the prior $D$ from collapsing and putting all the weight on a single source class. This is due to a high Wasserstein distance penalty that is incurred when the classes from source and target cannot be matched. Computationally, running an epoch of our algorithm takes a mere 0.17 seconds on our hardware for learning the transformation with Imagenet as the source and Pets as the target task with the ResNet-18 model (we ran the optimization for 2000 epochs).

---

**Algorithm for Task Transfer Analysis**

**Input:** Samples from the source $(\mathcal{Z}_S, \mathcal{Y}_S)$ and the target $(\mathcal{Z}_T, \mathcal{Y}_T)$ tasks
**Main:**

1. Sample $n_S$ points $(z_S^i, y_S^i) \sim (\mathcal{Z}_S, \mathcal{Y}_S)$ from the source task based on the class prior $D$ along with a random sample of $n_T$ points $(z_T^i, y_T^i) \sim (\mathcal{Z}_T, \mathcal{Y}_T)$ from the target.

2. Compute $(z_{S'''}^i, y_{S'''}^i) = (Az_S^i, \ \arg\max_y Be(y_S^i))$, for $i = 1, \cdots, n_S$.

3. Assign $\mathcal{Y}_{S'} = \mathcal{Y}_S$ and $\mathcal{Y}_{S''} = \mathcal{Y}_T$.

4. Compute the optimal coupling $\pi^*$ between the distributions $S'''$ and $T$ using type-1 Wasserstein distance via network simplex flow algorithm from POT (Flamary et al., 2021).

$$\min_{\pi \in \Pi(P_{S'''}, P_T)} \sum_{i,j} \pi_{ij} d((z_{S'''}^i, y_{S'''}^i), (z_T^j, y_T^j)) \text{ s.t. } \sum_j \pi_{ij} = \frac{1}{n_S} \ \forall i, \ \sum_i \pi_{ij} = \frac{1}{n_T} \ \forall j.$$

5. Using the coupling $\pi^*$, solve for $A, \bar{A}, B, D$ using mini-batch SGD

$$\min_{A, \bar{A}, B, D} \frac{1}{n_S} \sum_i \frac{D(y^i)}{P_S(y^i)} \ell(h_S(z_S^i), y^i) + H(\mathcal{Y}_{S''} | \mathcal{Y}_{S'}) + \sum_{i,j} \pi_{i,j}^* \left[ \tilde{d}((z_{S'''}^i, y_{S'''}^i), (z_T^j, y_T^j)) \right]$$

$$+ \|P_T(y) - BD\|_2^2 + (\|A\bar{A} - I\|_F + \|\bar{A}A - I\|_F).$$

6. Repeat 1 - 4 until convergence.

**Output:** Bound with the optimal transformations $A, \bar{A}, B, D$, and $W_d(P_{S'''}, P_T)$.

---

## 4.2 EFFECTIVENESS OF THE PROPOSED OPTIMIZATION

Here, we consider the transferability of a ResNet-18 model trained with Imagenet as the source and CIFAR10 as the target. We compare the cross-entropy loss on CIFAR10 obtained after linear fine-tuning with the transferability approximated via the bound in Theorem 3. We show how the different transformations affect our upper bound in Fig. 2 and show the advantage of learning the transformation by solving Eq. 3. We present an evaluation of two settings. In the first setting, we select data from 10 classes at random and 10 classes that are semantically related to the labels of CIFAR10 from Imagenet (Deng et al., 2009) (see App. D) and train a 10-way classifier for both ($h_S$). With these, we evaluate the bound, by fixing all transformations (**FixedAll:** $A$ is set to the Identity matrix, $B$ is a random permutation matrix, $D$ is set to the source prior), learning only $A$ (**LearnedA:** $A$ is optimized by solving Eq. 3 while $B, C$ are same as in FixedAll), and learning all the transformations (**LearnedAll**). The top part of Fig. 2 shows that for FixedAll, the presence of semantically related classes in the source data does not provide an advantage in terms of the bound compared to the presence of unrelated classes. This is attributed to $B$ being a random permutation which leads to matching between dissimilar classes from the two tasks. The bound becomes significantly better (decreases by $\sim 0.2$ from FixedAll to LearnedA) when the feature transform $A$ is learned due to the decrease in the

Wasserstein distance. Learning all the transformations produces the best upper bound with both random and semantically chosen classes, and the loss with semantically chosen classes is slightly better.

In the second setting, we select data from 20 classes from Imagenet by selecting either 20 random classes or 10 random and 10 semantically related classes. This setting allows us to analyze transferability when the source task has more classes than the target (for FixedAll and LearnedA, we fix $B$ to match two source classes entirely to a single target class.). In all cases, we observe that the upper bound becomes larger compared to our previous setting due to the increase of the re-weighted source loss as learning a 20-way classifier is more challenging than learning a 10-way classifier, especially with the Lipschitz constraint. Despite this, just learning the transform $A$ (LearnedA) produces a better upper bound. The bound is further improved by learning all the transformations. Fig. 5 (in the Appendix) shows that by learning $C$ the optimization prefers to retain the data from 10 of the 20 classes reducing the re-weighted source loss. Based on these insights we find that selecting the same number of classes as that present in the target task and learning only the transform $A$ keeping $B, C$ fixed to a random permutation matrix and prior of the source can achieve a smaller upper bound. Thus, we use this setup for all our other experiments. Additional experiments demonstrating the effectiveness of learning the transformations by solving Eq. 3, for different datasets, are present in App. C.1.2.

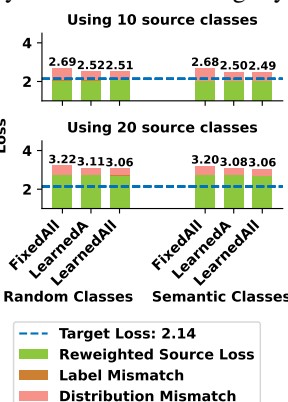

Figure 2: Transformations optimized using Eq. 3 produce a better bound than obtained by naively chosen transformations, primarily due to decreased distribution mismatch. The presence of semantically related source classes only has a limited impact on the bound compared to the presence of random classes.

### 4.3 IMPACT OF TASK RELATEDNESS ON TRANSFERABILITY

Here we evaluate how relatedness between the source task used for training the encoder and the downstream target task affects transferability. We consider a setup with convolutional neural networks trained using various character recognition tasks such as MNIST, Fashion-MNIST (FMNIST) (Xiao et al., 2017), KMNIST (Clanuwat et al., 2018), and USPS. We compute the pairwise transferability of these networks to other tasks. The results in Fig. 3 show that those target tasks achieve the best transfer-

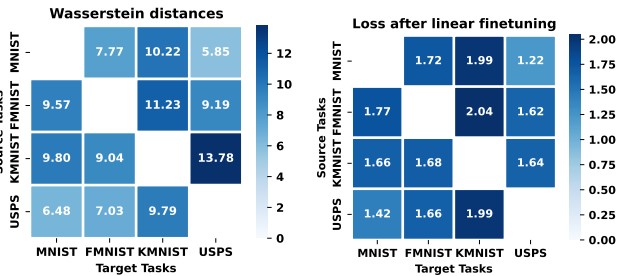

Figure 3: Task relatedness measured by $W_d(P_{S'''}, P_T)$ correlates strongly with transferability (i.e., target loss). Related tasks such as MNIST and USPS transfer better than unrelated tasks such as MNIST and KMNIST.

ability (small loss after linear fine-tuning) for which the source task (used for training the encoder) after transformation achieves the smallest residual Wasserstein distance, $W_d(P_{S'''}, P_T)$. Specifically, for the encoder trained on MNIST, USPS achieves the best transferability (1.22) and the smallest $W_d = 5.85$. A similar behavior is observed on MNIST for a USPS-trained encoder. This is intuitive as both datasets contain digits and the encoder trained for digits from the MNIST, correctly maps the corresponding digits from USPS. Moreover, the data/classifier of MNIST can be easily transformed to explain the performance of USPS as seen by small $W_d$. Similarly, when the source task cannot be transformed into the target task (indicated by high $W_d$) transferability suffers. Specifically, transferability is poor for KMNIST (consisting of Hiragana characters) since it is very different from the source tasks considered (MNIST, USPS, FMNIST) here for training the encoders, as indicated by high $W_d$. Moreover, the encoder trained with KMNIST also leads to poor transferability to other target tasks since they are unrelated to KMNIST. Thus, task relatedness and transferability are correlated. We explain the observed correlation as follows. As shown in Fig. 8 (in the Appendix), the bound is the source loss plus the $\tau W_d$. For the same source task, $W_d$ therefore explains the difference

of bounds for different target tasks. Since our bound is correlated well with transferability, $W_d$ is correlated with transferability. We also observe a similar correlation in sentence classification tasks in App. C.2.2: Figs. 9 and 12 again show that transferability improves when the two tasks are related. This conclusion is novel to our task transfer analysis and cannot be provided by SbTE methods since they explain transferability using only target tasks.

### 4.4 EVALUATION OF TRANSFERABILITY OF SOTA PRE-TRAINED CLASSIFIERS

Here we demonstrate the effectiveness of our bound and optimization at approximating transferability of pre-trained classifiers with architectures such as Vision Transformers (ViT) (Dosovitskiy et al., 2021), ResNet-18/50 (He et al., 2015) trained with different pretraining methods including supervised training, adversarial training (Salman et al., 2020), SimCLR (Chen et al., 2020), MoCo (He

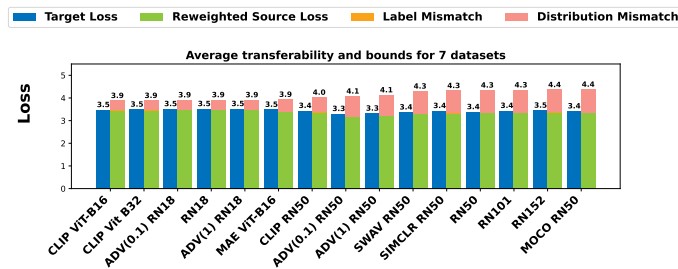

Figure 4: Transferability approximated via our bound has a small gap to empirical transferability for SOTA pre-trained models.

et al., 2020), SwAV (Caron et al., 2020), CLIP (Radford et al., 2021), and MAE (He et al., 2022). We consider a wide range of target datasets including, CIFAR10/100, Aircraft, Pets, and DTD whose details are in App. D. For this experiment, we consider the source task to be Imagenet (Deng et al., 2009) and use our optimization to learn transformations to approximate transferability. The results of this experiment in Fig. 4 and 10 (in the Appendix) show that regardless of the pretraining algorithm, the model architecture of the pre-trained classifier, and the target dataset, our bound achieves a small gap to empirical transferability (i.e. loss on the target task after linear fine-tuning). This shows that our upper bound is tight and useful for explaining the success of transfer learning in SOTA pre-trained classifiers. Moreover, in Fig. 11 (in the App. C.3.1) we show that our bound is strongly correlated even with the accuracy of the target task after linear fine-tuning. Our Fig. 4 and 10, show the contribution of individual terms in our upper bound and show that the label mismatch term has a small contribution when $B$ matches a single source class to a single target class (as opposed to matching a source class partially to two or more classes). From Fig. 10, we see that when the source and target tasks have fewer classes, learning $h_S$ is easier and the contribution of the re-weighted source loss term is relatively small but this loss increases as the number of classes increases. We also see that as the dimension of the representation space of the models increases (e.g., ResNet-18/ViT-B16 has a 512-dimensional representation space whereas ResNet50 models have 2048-dimensional space) estimating the Wasserstein distance becomes challenging and consequently minimizing it via $A$ also becomes challenging leading to an increased gap between empirical and approximated transferability. Nonetheless, our optimization finds a solution that consistently produces a small gap with empirical transferability demonstrating its effectiveness. In App. C.3.2, we present evaluation results on the sentence classification problem. Even there our upper bound achieves a small gap compared to empirical transferability.

## 5 CONCLUSION

We analyzed transfer learning with linear fine-tuning using a task transfer approach that works by transforming the source distribution and the classifier to match those of the target task. We proved a tight upper bound on transferability relating it to the performance of the model on the source task which can be decomposed into three terms, re-weighted source loss, label, and distribution mismatch. We also proposed an optimization problem to effectively learn the task transfer to approximate transferability and demonstrated its effectiveness at achieving a small gap to empirical transferability for SOTA models pre-trained with different architectures/training methods. Our results highlight the importance of relatedness between source and downstream tasks, measured by the relative ease with which the source task can be transformed to the target to achieve high transferability. Extending this analysis to full fine-tuning and different loss functions are promising directions for future research.

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

# Appendix

We present the missing proofs of the theoretical results from Sec. 3 along with justifications for the classifiers $(h_{S'}, h_{S''}, h_{S'''})$ as Corollaries in Appendix A followed by related work on learning in the presence of distribution shift with the same feature and label space in Appendix B. This is followed by additional experimental results including NLP classification tasks with large pretrained models in Appendix C. We conclude in Appendix D with details of the experiments and datasets used.

## A PROOFS FOR SEC. 3

### A.1 ANALYSIS OF THE TASK TRANSFER MODEL (SEC. 3.1)

#### A.1.1 PRIOR TRANSFORM $(S \to S')$

**Lemma 1.** *Let* $C := \left[\frac{P_{S'}(y)}{P_S(y)}\right]_{y=1}^{|\mathcal{Y}_S|}$ *be a vector of probability ratios and the classifier* $h_{S'}(z) := h_S(z)$, *then* $\mathbb{E}_{P_{S'}(z,y)}[\ell(h_{S'}(z), y)] = \mathbb{E}_{P_S(z,y)}[C(y)\ell(h_S(z), y)]$, *for any loss function* $\ell$.

*Proof.*

$$\mathbb{E}_{P_{S'}(z,y)}[\ell(h_{S'}(z), y)] = \mathbb{E}_{P_{S'}(z,y)}[\ell(h_S(z), y)] = \sum_{y \in \mathcal{Y}_S} P_{S'}(y)\mathbb{E}_{P_{S'}(z|y)}[\ell(h_S(z), y)]$$

$$= \sum_{y \in \mathcal{Y}_S} \frac{P_S(y)}{P_S(y)} P_{S'}(y)\mathbb{E}_{P_{S'}(z|y)}[\ell(h_S(z), y)] = \sum_{y \in \mathcal{Y}_S} P_S(y)\mathbb{E}_{P_{S'}(z|y)}[C(y)\ell(h_S(z), y)]$$

$$= \sum_{y \in \mathcal{Y}_S} P_S(y)\mathbb{E}_{P_S(z|y)}[C(y)\ell(h_S(z), y)] \quad \text{(since } P_S(z|y) = P_{S'}(z|y) \text{ by construction)}$$

$$= \mathbb{E}_{P_S(z,y)}[C(y)\ell(h_S(z), y)].$$

$\square$

#### A.1.2 LABEL TRANSFORM $(S' \to S'')$

**Lemma 2.** *Let* $B$ *be a* $|\mathcal{Y}_S| \times |\mathcal{Y}_T|$ *matrix with* $B_{ij} = P(y_{S''} = i|y_{S'} = j)$ *and* $h_{S''}(z) := Bh_{S'}(z)$ *and* $\ell$ *be the cross-entropy loss. Then,* $\mathbb{E}_{P_{S''}(z,y)}[\ell(h_{S''}(z), y)] \leq \mathbb{E}_{P_{S'}(z,y)}[\ell(h_{S'}(z), y)] + H(\mathcal{Y}_{S''}|\mathcal{Y}_{S'})$, *where* $H(\mathcal{Y}_{S''}|\mathcal{Y}_{S'}) := [-\sum_{y_{S'} \in \mathcal{Y}_{S'}} \sum_{y_{S''} \in \mathcal{Y}_{S''}} P_{S'}(y_{S'})B_{y_{S''},y_{S'}} \log(B_{y_{S''},y_{S'}})]$ *denotes the conditional entropy.*

*Proof.* Note that $P(z) := P_{S'}(z) = P_{S''}(z)$ by construction.

$$\mathbb{E}_{P_{S''}(z,y)}[\ell(h_{S''}(z), y)] = \mathbb{E}_{P(z,y'')}[\ell(h_{S''}(z), y'')]$$

$$= \mathbb{E}_{P(z)}\mathbb{E}_{P(y''|z)}[\ell(h_{S''}(z), y'')] = \mathbb{E}_{P(z)}\sum_{y''}\sum_{y'} P(y'', y'|z)[\ell(h_{S''}(z), y'')] \quad \text{(since } y' \in \mathcal{Y}_{S'})$$

$$= \mathbb{E}_{P(z)}\mathbb{E}_{P(y'',y'|z)}[\ell(h_{S''}(z), y'')]$$

$$= \mathbb{E}_{P(z)}\mathbb{E}_{P(y'|z)}\mathbb{E}_{P(y''|y')}[\ell(h_{S''}(z), y'')] \quad \text{(since } P(y''|y', z) = P(y''|y'))$$

$$= \mathbb{E}_{P(z)}\mathbb{E}_{P(y'|z)}[\sum_{y'' \in \mathcal{Y}_{S''}} \ell(h_{S''}(z), y'')B_{y'',y'}] \quad \text{(since } B_{y'',y'} = P(y''|y'))$$

$$= \mathbb{E}_{P(z,y')}[\sum_{y'' \in \mathcal{Y}_{S''}} \ell(Bh_{S'}(z), y'')B_{y'',y'}].$$

Since the loss $\ell$ is the cross-entropy loss, we have

$$\ell(Bh_{S'}(z), y'') = \ell(\sum_{j \in \mathcal{Y}_{S'}} B_{y'',j}h_{S'}^j(z)) = -\log(\sum_{j \in \mathcal{Y}_{S'}} B_{y'',j}h_{S'}^j(z))$$

$$\leq -\log(B_{y'',y'}h_{S'}^{y'}(z)) = -\log(B_{y'',y'}) - \log(h_{S'}^{y'}(z)).$$

Therefore, we have

$$
\begin{aligned}
& \mathbb{E}_{P_{S''}(z,y)}[\ell(h_{S''}(z), y)] \\
=\ & \mathbb{E}_{P(z,y')}\big[ \sum_{y'' \in \mathcal{Y}_{S''}} \ell(Bh_{S'}(z), y'') B_{y'',y'} \big] \\
\leq\ & -\mathbb{E}_{P(z,y')}\big[ \sum_{y'' \in \mathcal{Y}_{S''}} B_{y'',y'} \big( \log(B_{y'',y'}) + \log(h_{S'}^{y'}(z)) \big) \big] \\
=\ & -\mathbb{E}_{P(z,y')}\big[ \sum_{y'' \in \mathcal{Y}_{S''}} B_{y'',y'} \log(B_{y'',y'}) \big] - \mathbb{E}_{P(z,y')}\big[ \sum_{y'' \in \mathcal{Y}_{S''}} B_{y'',y'} \log(h_{S'}^{y'}(z)) \big] \\
=\ & -\mathbb{E}_{P(z,y')}\big[ \sum_{y'' \in \mathcal{Y}_{S''}} B_{y'',y'} \log(B_{y'',y'}) \big] + \mathbb{E}_{P(z,y')}\big[ -\log(h_{S'}^{y'}(z)) \sum_{y'' \in \mathcal{Y}_{S''}} B_{y'',y'} \big] \\
=\ & -\mathbb{E}_{P(z,y')}\big[ \sum_{y'' \in \mathcal{Y}_{S''}} B_{y'',y'} \log(B_{y'',y'}) \big] + \mathbb{E}_{P(z,y')}[ -\log(h_{S'}^{y'}(z)) ] \\
=\ & \mathbb{E}_{P(z,y')}\big[ -\sum_{y'' \in \mathcal{Y}_{S''}} B_{y'',y'} \log(B_{y'',y'}) \big] + \mathbb{E}_{P(z,y')}[ \ell(h_{S'}(z), y') ] \\
=\ & \mathbb{E}_{P(y')}\mathbb{E}_{P_{S'}(z|y')}\big[ -\sum_{y'' \in \mathcal{Y}_{S''}} B_{y'',y'} \log(B_{y'',y'}) \big] + \mathbb{E}_{P(z,y')}[ \ell(h_{S'}(z), y') ] \\
=\ & \big[ -\sum_{y' \in \mathcal{Y}_{S'}} \sum_{y'' \in \mathcal{Y}_{S''}} P(y') B_{y'',y'} \log(B_{y'',y'}) \big] + \mathbb{E}_{P(z,y')}[ \ell(h_{S'}(z), y') ] \\
=\ & H(\mathcal{Y}_{S''}|\mathcal{Y}_{S'}) + \mathbb{E}_{P_{S'}(z,y)}[ \ell(h_{S'}(z), y) ].
\end{aligned}
$$

$\square$

Corollary 2 below, shows the conditions under which the optimal softmax classifier for the domain $S'$ remains optimal for the domain $S''$, justifying our choice of classifier change from $S'$ to $S''$.

**Corollary 2.** *Let $e$ be one-hot encoding of the labels, $|\mathcal{Y}_{S''}| = |\mathcal{Y}_{S'}|$, $B$ be a $|\mathcal{Y}_S| \times |\mathcal{Y}_T|$ permutation matrix and $h_{S'}$ be the optimal softmax classifier for $S'$ and $y_{S''} := \sigma(y_{S'}) := \arg\max_{y \in \mathcal{Y}_{S''}} (Be(y_{S'}))_y$ then under the assumptions of Lemma 2, $h_{S''}(z) := Bh_{S'}(z)$ is the optimal softmax classifier for $S''$.*

*Proof.* Since $y_{S''} := \sigma(y_{S'}) := \arg\max_{y \in \mathcal{Y}_T} (Be(y_{S'}))_y$ we have

$$
\begin{aligned}
\mathbb{E}_{P_{S''}}[\ell(h_{S''}(z), y_{S''})] &= \mathbb{E}_{P(z,y'')}[\ell(Bh_{S'}(z), y'')] = \sum_{y'' \in \mathcal{Y}_{S''}} P(y'') \mathbb{E}_{P(z|y'')}[\ell(Bh_{S'}(z), y'')] \\
&= \sum_{y' \in \mathcal{Y}_{S'}} P(\sigma(y')) \mathbb{E}_{P(z|\sigma(y'))}[\ell(Bh_{S'}(z), \sigma(y'))] \\
&= \sum_{y' \in \mathcal{Y}_{S'}} P(y') \mathbb{E}_{P(z|y')}[\ell(h_{S'}(z), y')] = \mathbb{E}_{P_{S'}}[\ell(h_{S'}(z), y_{S'})].
\end{aligned}
$$

The second last equality follows due to the symmetry of cross-entropy loss, i.e., $\ell(h, y) = -\log h_y = -\log Bh_{\sigma(y)} = \ell(Bh, \sigma(y))$.

Since $\min_{h_{S''}} \mathbb{E}_{P_{S''}}[\ell(h_{S''}(z), y_{S''})] = \min_{h_{S'}} \mathbb{E}_{P_{S'}}[\ell(h_{S'}(z), y_{S'})]$ and $h_{S'}$ is optimal for $S'$ we have $h_{S''}(z) := Bh_{S'}(z)$ is the optimal softmax classifier for $S''$. $\square$

### A.1.3 FEATURE TRANSFORM ($S'' \to S'''$)

**Lemma 3.** *Let $A : \mathcal{Z} \to \mathcal{Z}$ be an invertible linear map of features and the classifier $h_{S'''}(z_{S'''}) := h_{S''}(A^{-1}(z_{S'''}))$. Then $\mathbb{E}_{P_{S'''}(z,y)}[\ell(h_{S'''}(z), y)] = \mathbb{E}_{P_{S''}(z,y)}[\ell(h_{S''}(z), y)]$ for any loss $\ell$.*

*Proof.* $\mathbb{E}_{P_{S'''}(z,y)}[\ell(h_{S'''}(z), y)] = \mathbb{E}_{P_{S'''}(z,y)}[\ell(h_{S''}(A^{-1}(z)), y)] = \mathbb{E}_{P_{S''}(z,y)}[\ell(h_{S''}(z), y)]$. $\square$

Our Corollary 3 below shows that the optimal softmax classifier for domain $S''$ remains optimal for domain $S'''$ too.

**Corollary 3.** *Let $h_{S''}$ be the optimal softmax classifier in domain $S''$ then under the assumptions of Lemma 3, $h_{S'''}(x_{S'''}) = h_{S''}(A^{-1}(x_{S'''}))$ is the optimal softmax classifier in domain $S'''$.*

*Proof.* When $h_{S'''}(x_{S'''}) = h_{S''}(A^{-1}(x_{S'''}))$, $\min_{h_{S''}} \mathbb{E}_{P_{S''}(z,y)}[\ell(h_{S''}(z), y)] = \min_{h_{S'''}} \mathbb{E}_{P_{S'''}(z,y)}[\ell(h_{S'''}(z), y)]$ by Lemma 3, hence if $h_{S''}$ is optimal for $S''$ then so is $h_{S'''}$ for the domain $S'''$. $\qquad\square$

### A.1.4 THREE TRANSFORMATIONS COMBINED $(S \to S''')$

**Theorem 1.** *Under the assumptions of Lemmas 1, 2, 3 we have*

$$\mathbb{E}_{P_{S'''}(z,y)}[\ell(h_{S'''}(z), y)] \leq \mathbb{E}_{P_S(z,y)}[C(y)\ell(h_S(z), y)] + H(\mathcal{Y}_{S''}|\mathcal{Y}_{S'}).$$

*Proof.*

$$
\begin{aligned}
\mathbb{E}_{P_{S'''}(z,y)}[\ell(h_{S'''}(z), y)] &= \mathbb{E}_{P_{S''}(z,y)}[\ell(h_{S''}(z), y)] \text{ (Lemma 3)} \\
&\leq \mathbb{E}_{P_{S'}(z,y)}[\ell(h_{S'}(z), y)] + H(\mathcal{Y}_{S''}|\mathcal{Y}_{S'}) \text{ (Lemma 2)} \\
&= \mathbb{E}_{P_S(z,y)}[C(y)\ell(h_S(z), y)] + H(\mathcal{Y}_{S''}|\mathcal{Y}_{S'}) \text{ (Lemma 1).}
\end{aligned}
$$

$\qquad\square$

**Corollary 1.** *Let $e$ be one-hot encoding of the labels, $|Y_{S'''}| = |Y_S|$, $B : \Delta_{S'} \to \Delta_{S''}$ be a permutation matrix and $y_{S''} := \sigma(y_{S'}) := \arg\max_{y \in \mathcal{Y}_{S''}}(Be(y_{S'}))_y$ then under the assumptions of Lemmas 1, 2, and 3 we have $\mathbb{E}_{P_{S'''}(z,y)}[\ell(h_{S'''}(z), y)] = \mathbb{E}_{P_S(z,y)}[C(y)\ell(h_S(z), y)]$.*

*Proof.* Since $y_{S''} := \sigma(y_{S'}) := \arg\max_{y \in \mathcal{Y}_T}(Be(y_{S'}))_y$ we have

$$
\begin{aligned}
\mathbb{E}_{P_{S''}}[\ell(h_{S''}(z), y_{S''})] &= \mathbb{E}_{P(z,y'')}[\ell(Bh_{S'}(z), y'')] = \sum_{y'' \in \mathcal{Y}_{S''}} P(y'')\mathbb{E}_{P(z|y'')}[\ell(Bh_{S'}(z), y'')] \\
&= \sum_{y' \in \mathcal{Y}_{S'}} P(\sigma(y'))\mathbb{E}_{P(z|\sigma(y'))}[\ell(Bh_{S'}(z), \sigma(y'))] \\
&= \sum_{y' \in \mathcal{Y}_{S'}} P(y')\mathbb{E}_{P(z|y')}[\ell(h_{S'}(z), y')] = \mathbb{E}_{P_{S'}(z,y)}[\ell(h_{S'}(z), y)].
\end{aligned}
$$

The second last equality follows due to the symmetry of cross-entropy loss, i.e., $\ell(h, y) = -\log h_y = -\log Bh_{\sigma(y)} = \ell(Bh, \sigma(y))$.

Therefore, we have

$$
\begin{aligned}
\mathbb{E}_{P_{S'''}(z,y)}[\ell(h_{S'''}(z), y)] &= \mathbb{E}_{P_{S''}(z,y)}[\ell(h_{S''}(z), y)] \text{ (Lemma 3)} \\
&= \mathbb{E}_{P_{S'}(z,y)}[\ell(h_{S'}(z), y)] \text{ (from above)} \\
&= \mathbb{E}_{P_S(z,y)}[C(y)\ell(h_S(z), y)] \text{ (Lemma 1).}
\end{aligned}
$$

$\qquad\square$

## A.2 DISTRIBUTION MISMATCH BETWEEN $S'''$ AND $T$ (SEC. 3.2)

**Lemma 4.** *Let $R$ and $Q$ be two distributions on $\mathcal{Z} \times \mathcal{Y}$ with the same prior $P_R(y = i) = P_Q(y = i) = P(y = i)$. With the base distance $d$ defined as in Eq. 1, we have $W_d(P_R, P_Q) = \sum_y P(y)W_{\|\cdot\|_2}(P_R(z|y), P_Q(z|y))$.*

*Proof.* Let $\omega_y^*$ denote the optimal coupling for the conditional distributions $(P_R(z|y), P_Q(z|y))$ for $y \in \mathcal{Y}$ and $\pi^*$ denote the the optimal coupling for the joint distributions $(P_R(z, y), P_Q(z, y))$. Then, under the definition of our base distance $d$, $\pi^*((z, y), (z', y')) = 0$ when $y \neq y'$ i.e. no mass from the distribution $R$ belonging to class $y$ can be moved to the classes $y' \neq y$ of the distribution $Q$ when the class priors of $R$ and $Q$ are the same. Moreover, since $\sum_{ij}(\omega_y^*)_{ij} = 1$ and

$\sum_{\{i,j:y_i=y'_j=k\}} \pi^*_{ij} = P(y = k)$ for $k \in \mathcal{Y}$ we have $\pi^*((z, y), (z', y')) = \omega^*_y(z, z')P(y)1_{y=y'}$ for every $y, y' \in \mathcal{Y}$.

Then, we can show that the total Wasserstein distance between the joint distributions can be expressed as the sum of conditional Wasserstein distances, as follows

$$
\begin{aligned}
&W_d(P_R(z, y), P_Q(z, y)) \\
=\ & \sum_{y,y'} \int \pi^*((z, y), (z', y'))d((z, y), (z', y'))dzdz' \\
=\ & \sum_{y,y'} \int \pi^*((z, y), (z', y'))(\|z - z'\|_2 + \infty \cdot 1_{y \neq y'})dzdz' \\
=\ & \sum_{y,y'} \int \omega^*_y(z, z')P(y)1_{y=y'}(\|z - z'\|_2 + \infty \cdot 1_{y \neq y'})dzdz' \\
=\ & \sum_{y,y'} \int \omega^*_y(z, z')P(y)1_{y=y'}\|z - z'\|_2 dzdz' \quad (\text{since } 1_{y=y'} \cdot 1_{y \neq y'} = 0) \\
=\ & \sum_{y,y'} P(y)1_{y=y'} \int \omega^*_y(z, z')\|z - z'\|_2 dzdz' \\
=\ & \sum_y P(y) \int \omega^*_y(z, z')\|z - z'\|_2 dzdz' \\
=\ & \sum_y P(y)W_{\|\cdot\|_2}(P_R(z|y), P_Q(z|y)).
\end{aligned}
$$

$\square$

**Theorem 2.** *Let the distributions $T$ and $S'''$ be defined on the same domain $\mathcal{Z} \times \mathcal{Y}$ and assumption 1 holds, then $\mathbb{E}_{P_T(z,y)}[\ell(h(z), y)] - \mathbb{E}_{P_{S'''}(z,y)}[\ell(h(z), y)] \leq \underbrace{\tau\, W_d(P_{S'''}, P_T)}_{\textit{Distribution mismatch}}$, with $d$ as in Eq. 1.*

*Proof.*

$$
\begin{aligned}
&\mathbb{E}_{P_T(z,y)}[\ell(h(z), y)] - \mathbb{E}_{P_{S'''}(z,y)}[\ell(h(z), y)] \\
=\ & \mathbb{E}_{P_T(y)}\mathbb{E}_{P_T(z|y)}[\ell(h(z), y)] - \mathbb{E}_{P_{S'''}(y)}\mathbb{E}_{P_{S'''}(z|y)}[\ell(h(z), y)] \\
=\ & \mathbb{E}_{P_T(y)}[\mathbb{E}_{P_T(z|y)}[\ell(h(z), y)] - \mathbb{E}_{P_{S'''}(z|y)}[\ell(h(z), y)]] \ (\text{since } P_T(y) = P_{S'''}(y)) \\
\leq\ & \mathbb{E}_{P_T(y)}[\sup_{\ell' \circ h' \in \tau - Lipschitz} \mathbb{E}_{P_T(z|y)}[\ell'(h'(z), y)] - \mathbb{E}_{P_{S'''}(z|y)}[\ell'(h'(z), y)]] \\
=\ & \mathbb{E}_{P_T(y)}[\tau\, W_{\|\cdot\|_2}(P_T(z|y), P_{S'''}(z|y))] \ (\text{Kantorovich} - \text{Rubinstein duality}) \\
=\ & \tau\, W_d(P_{S'''}, P_T) \ (\text{Lemma 4}).
\end{aligned}
$$

$\square$

## A.3 FINAL BOUND (SEC. 3.3)

**Theorem 3.** *Let $\ell$ be the cross entropy loss then under the assumptions of Theorems 1 and 2 we have,*

$$
\mathbb{E}_{P_T(z,y)}[\ell(h_T(z), y)] \leq \mathbb{E}_{P_S(z,y)}[C(y)\ell(h_S(z), y)] + H(\mathcal{Y}_{S''}|\mathcal{Y}_{S'}) + \tau\, W_d(P_{S'''}, P_T)
$$

*Proof.* Let $\ell \circ h_T$, $\ell \circ h_S$, and $\ell \circ h_{S'''}$ be $\tau-$Lipschitz (from Assumption 1).

$$
\begin{aligned}
&\mathbb{E}_{P_T(z,y)}[\ell(h_T(z), y)] \leq \mathbb{E}_{P_T(z,y)}[\ell(h_{S'''}(z), y)] \ (\text{Optimality difference}) \\
\leq\ & \mathbb{E}_{P_{S'''}(z,y)}[\ell(h_{S'''}(z), y)] + \tau\, W_d(P_{S'''}, P_T) \ (\text{Theorem 2}) \\
\leq\ & \mathbb{E}_{P_S(z,y)}[C(y)\ell(h_S(z), y)] + H(\mathcal{Y}_{S''}|\mathcal{Y}_{S'}) + \tau\, W_d(P_{S'''}, P_T) \ (\text{Theorem 1}).
\end{aligned}
$$

$\square$

In our experiments, we enforce the $\tau-$Lipschitz constraint for $\ell \circ h_S$ and $\ell \circ h_T$ and verify that the Lipschitz constant of $\ell \circ h_{S'''}$ remains close to $\tau$ within tolerance.

### A.4 EXTENSION TO NON-LINEAR CLASSIFIERS AND NON-LINEAR TRANSFORMATIONS

To extend our analysis, we allow $A : \mathcal{Z} \to \mathcal{Z}$ to be a non-linear map and the classifiers $h \in \mathcal{H}_{\text{non\_linear}}$ to be also non-linear (such as multi-layer perceptron). In addition to the Assumption 1 that $\ell \circ h_{S'''}$ is $\tau-$Lipschitz, we also require that $h_{S'''}$ belongs to the same class $\mathcal{H}_{\text{non\_linear}}$ as $h_T$ and $h_S$ for any $A$. For example, this holds when $h$ is linear and $A$ is also linear. For another, this holds when $h$ is a multilayer perceptron and $A$ is linear. With the additional assumption, all proofs of this paper are in fact general and work for any linear/non-linear transformation of the feature space. Thus, Theorem 1, Theorem 2 and Theorem 3 hold for non-linear classifiers as well. With these extensions, our bounds can be used to explain transferability even with non-linear classifiers.

## B ADDITIONAL RELATED WORK

**Distributional divergence-based analyses of learning with distribution shifts (under same feature and label sets):** Here we review some of the previous works that analyzed the problem of learning under distribution shifts in terms of distributional divergences such as the Wasserstein distance. These analyses apply when the feature and label spaces remain the same between the original and the shifted distribution.

Early works (Ben-David et al., 2007; Shen et al., 2018; Mansour et al., 2009) showed that the performance on a shifted distribution (target domain) can be estimated in terms of the performance of the source domain and the distance between the two domains' marginal distributions and labeling functions. Specifically, (Ben-David et al., 2007) showed that that

$$\mathcal{E}_T(h, f_T) \leq \mathcal{E}_S(h, f_S) + d_1(P_S, P_T) + \min\{\mathbb{E}_{P_S}[|f_S(z) - f_T(z)|], \mathbb{E}_{\mathcal{D}_T}[|f_S(z) - f_T(z)|]\},$$

where $d_1$ denotes the total variation distance, $f : \mathcal{Z} \to [0, 1]$ denotes the labeling function, $h : \mathcal{Z} \to \{0, 1\}$ denotes the hypothesis and $\mathcal{E}_P(h, f) := \mathbb{E}_{z \sim P}[|h(z) - f(z)|]$ denote the risk of the hypothesis $h$. A follow up work (Shen et al., 2018), showed a similar result using type-1 Wasserstein distance for all $K-$Lipschitz continuous hypotheses i.e.,

$$\mathcal{E}_T(h, f_T) \leq \mathcal{E}_S(h, f_S) + 2K \cdot W_1(P_S, P_T) + \lambda,$$

where $\lambda$ is the combined error of the ideal hypothesis $h^*$ that minimizes the combined error $\mathcal{E}_S(h, f_S) + \mathcal{E}_T(h, f_T)$. Another recent work (Le et al., 2021) used a target transformation-based approach and Wasserstein distance to quantify learning in the presence of data and label shifts. Other works (Kumar et al., 2022; Sehwag et al., 2021) also presented an analysis based on Wasserstein distance to understand how the accuracy of smoothed classifies and robustness change in the presence of distribution shifts. Compared to these works the bound proposed in Theorem 2 considers cross-entropy loss (which is a popular choice of the loss function in the classification setting) and uses a joint feature and labels Wasserstein distance rather than only marginal Wasserstein distance. These differences make the bound proposed in Theorem 2 useful in the analysis of transfer learning than those proposed in previous works when we have access to labeled target domain data.

**Comparison with Tran et al. (2019):** The closest work to ours is that of (Tran et al., 2019), which also proposed an upper bound on transferability in terms of the performance of the source task. However, the bound is proposed in a restrictive setting when the source and target tasks have the same features but different labels (i.e. same images labeled differently between the source and target tasks). In this setting, Tran et al. (2019), showed that transferability is upper bounded by loss of the source classifier on the source task and the conditional entropy (CE) of the label sets of the two tasks. We significantly extend this analysis to general source and target tasks which is the most commonly used setting in practice (for e.g., our analysis allows us to study transfer learning from Imagenet with 1000 classes to CIFAR-100 with 100 unrelated classes, where source and target tasks do not have the same images). In this setting, our main result in Theorem 3, shows that transferability involves additional terms such as the distribution mismatch term (in the form of Wasserstein distance), the prior mismatch term (in the form of re-weighted source loss) and the conditional entropy between the label sets. Moreover, the bound proposed by Tran et al. (2019) is a special case of our bound with $C$

being the vector of all ones (no prior change) and $A$ being Identity (data distributions of source and target are the same).

**Comparison with score-based transferability estimation (SbTE) methods:** The problem of transferability estimation has gained a lot of attention recently, especially due to the availability of a larger number of pretrained models. As a result, a large body of work (Bao et al., 2019; Nguyen et al., 2020; Huang et al., 2022; You et al., 2021; Tan et al., 2021; Shao et al., 2022; Ding et al., 2022; Dwivedi & Roig, 2019; Dwivedi et al., 2020) focuses on the problem of pretrained model selection where the goal is to find the best pretrained classifier from a model zoo that will achieve the highest transferability to a particular downstream task. The main challenge of this problem is to be able to estimate transferability in a way that is more efficient than actually fine-tuning the pretrained models on the downstream tasks. To this end, researchers have proposed several scores that have been shown to be correlated with the accuracy of the models after fine-tuning them on the target task. However, unlike our work, the goal of SbTE works is not to propose a universal bound or identify terms that govern transferability from source to target tasks.

Moreover, while the scores proposed in SbTE works correlate well with transferability, they are only meaningful in a relative sense. Concretely, a score of 1 (e.g., for LogMe [46]) on a CIFAR-100 task for a particular model does not indicate whether transferability is good or bad and requires comparison with scores of other pre-trained models on the same target task. On the other hand, our upper bound directly approximates transferability, e.g., an upper bound of 1 on the CIFAR-100 task for a model implies that transfer learning will incur an average cross-entropy loss of less than 1 implying a highly accurate transfer. Our results in Figs. 4, 10 and 12 attest that our upper bound is indeed a good estimate of the transferability.

Another disadvantage of the scores proposed in these works is that they cannot be compared across target tasks, unlike our upper bound. As observed from Fig. 4 of LogMe [46], scores for CIFAR-10 are lower than scores for CIFAR-100 on the same pretrained model, but, the transferability to CIFAR-10 is better than that to CIFAR-100. On the other hand, from our Fig. 10, the upper bounds on CIFAR-10 are lower than those of CIFAR-100 implying better transferability of classifiers pretrained on Imagenet to CIFAR-10. Thus, our work is more suitable for estimating the absolute performance on various target tasks given a particular pretrained classifier. Lastly, our upper bound also encodes how task-relatedness affects transferability (Fig. 3 and 9) which cannot be explained by SbTE approaches since transferability is not studied in comparison to the performance on the source task in these works.

Thus, the goal of our work differs fundamentally from SbTE approaches since both serve different purposes. Specifically, if the goal is pretrained model selection then SbTE approaches are more suitable but if the goal is to analyze the transferability of models to downstream tasks, a rigorous analysis as proposed in our work is necessary.

For completeness, we present a comparison of using our upper bound for the problem of pre-trained model selection in App. C.4 and show that it achieves a competitive performance compared to various SbTE metrics.

## C  ADDITIONAL EXPERIMENTS

### C.1  ADDITIONAL RESULTS FOR THE EFFECTIVENESS OF THE PROPOSED OPTIMIZATION (SEC. 4.2)

#### C.1.1  VISUALIZATION OF THE TRANSFORMED DATA VIA T-SNE FOR VARIOUS SETTINGS IN SEC. 4.2

In this section, we use the setting considered in Sec. 4.2 where we consider 20 randomly selected classes from Imagenet as the source and consider the transfer to CIFAR-10. We plot the results of using different transformations using t-SNE to show how various transformations affect the upper bound in Theorem 3. Our results in Fig. 5(left) show that when no transformations are learned (FixedAll), the 20 random source classes do not overlap with the 10 target classes leading to an increased Wasserstein distance which in turn leads to a larger upper bound. By learning the transformation $A$ (LearnedA), Fig. 5(center) shows a significantly better alignment between the classes of the source and target which leads to a decreased Wasserstein distance and hence a tighter

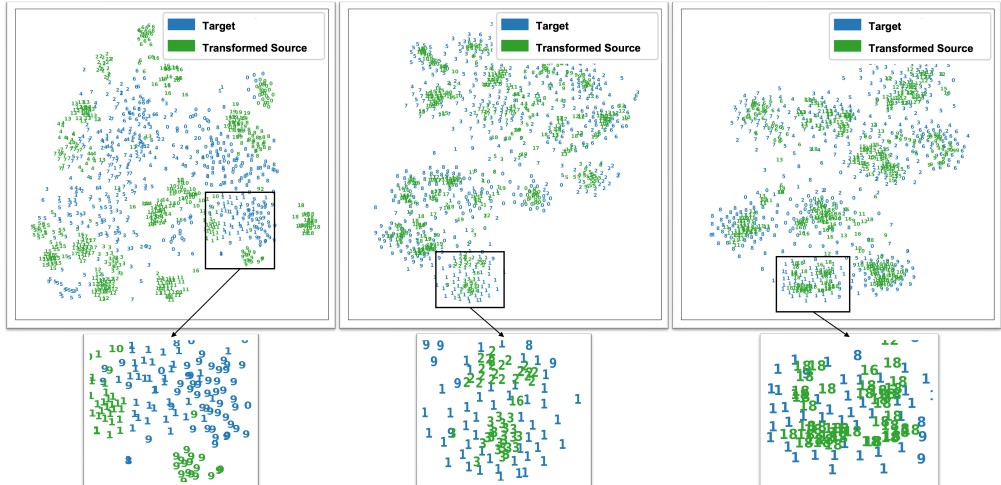

Figure 5: (Best viewed in color.) t-SNE visualizations of the effect of various transformations on the bound in Theorem 3 when data from 20 randomly selected classes from Imagenet are used to transfer to CIFAR-10. When all transformations are fixed (FixedAll, left) the distance between the distribution $S'''$ (transformed source) and $T$ is high explaining the large upper bound. Learning just the transformation $A$ using the algorithm proposed in Sec. 4.1 significantly reduces the distance between $S'''$ and $T$ leading to a tighter upper bound (center). Learning all the transformations further improves the matching (right). Especially, learning $B$ and $D$ change the class priors of the source so that the same number of classes from the source are used for matching as those present in the target. This is evident from the right plot where only 10 unique source clusters are visible compared to 20 in the center plot, with fixed D. Moreover, the zoomed-in portion shows that for the center figure two classes from the source (green 2 and 3) match with class 1 (blue) of the target whereas a single class from the source (green 18) matches class 1 (blue) of the target in the right figure.

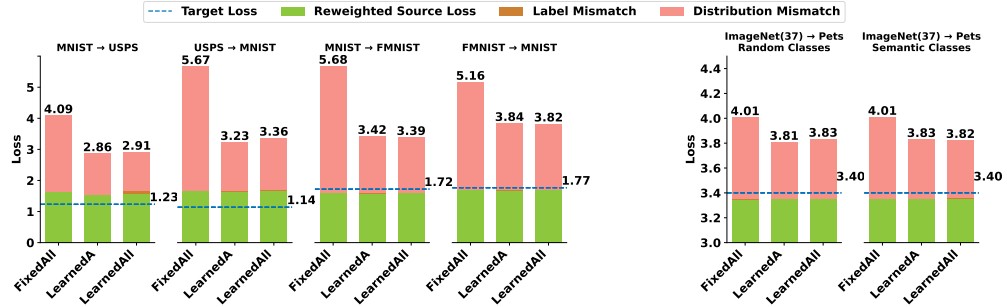

Figure 6: Comparison of using various transformations on the upper bound for different datasets. For all datasets learning the transformation by solving the optimization problem in Eq. 3 leads to a significantly tighter upper bound on the loss incurred after linear fine-tuning.

upper bound. Moreover, by learning all the transformations (LearnedAll), Fig. 5(right) shows that not only do the distributions align well but also the prior of the source is changed to only keep 10 source classes to match the prior of the target distribution providing a further improvement in the upper bound. This clearly shows the effectiveness of our proposed optimization algorithm in learning various transformations to minimize the upper bound.

### C.1.2 ADDITIONAL EXPERIMENTS FOR TRANSFERRING IN DIFFERENT SETTINGS/DATASETS

Here we extend the experiment presented in Sec. 4.2 to the Pets dataset. Specifically, we select data from 37 random and semantically related (list in App. D) source classes from Imagenet and use them to transfer to the Pets dataset. Results for this experiment are present in Fig. 6 (2 right-most plots).

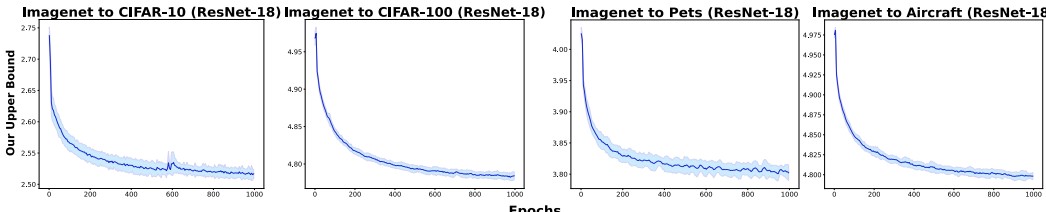

Figure 7: Reduction of the proposed upper bound is shown as transformations are learned by solving the optimization problem in Eq. 3. After 600 epochs, the upper bound stabilizes showing the convergence of the optimization problem. Each subplot shows the effect of learning the transformation parameters for the transfer learning task with Imagenet as the source and ResNet-18 (trained in a supervised way) for different target tasks. The solid line is the average after 5 random restarts and the shaded portion shows their standard deviation.

Consistent with the results presented in the main paper, we find that learning just the transformation $A$ produces a significantly better upper bound than when all transformations are fixed. Moreover, learning all the transformations produces a similar or slightly better result than learning only $A$ in this setting. Lastly, the presence of semantically related or random classes in the source does not produce a significant difference in terms of the bound.

Next, we evaluate the effectiveness of the optimization on datasets such as MNIST, FMNIST, and USPS. For these datasets, we first train a convolutional neural network model on the data from the source task and then perform linear fine-tuning using the data from the target task. Similar to the previous experiments, the results in Fig. 6 (first 4 plots) show that when the transformations are fixed, the gap between the loss on the target data after linear fine-tuning and the upper bound is large. Learning the transformations by solving the proposed optimization in Eq. 3 reduces this gap significantly. For the experiment with the source as MNIST and target as USPS and vice-versa, we additionally compare our results to a setting where only $A$ is learned and $B$ is set to an identity matrix (rather than a permutation matrix, as used in LearnedA setting). This matrix $B$ contains the correct matching between the labels of the source and target. We find that the upper bound obtained when $B$ is fixed to identity is only marginally better than the case when $B$ is a random permutation or is learned through solving our optimization problem. Specifically, with $B$ fixed to identity the upper bound improves by 0.15 for the MNIST→USPS task (from 2.91 with LearnedAll to 2.76) and by 0.3 for the USPS→MNIST task (from 3.36 with LearnedAll to 3.06). As expected the primary reason for the decrease in the upper bound comes from the reduced Wasserstein distance $W_d(S''', T)$ and the label mismatch term. While the upper bound improves slightly when the ideal matching between the labels is known, such a mapping may not be known when the labels of the tasks are not related such as for FMNIST and MNIST. Moreover, due to the difficulty of the optimization problem (different label associations producing similar upper bounds) recovering the true association between the labels could be hard. Similar to (Alvarez-Melis & Fusi, 2020), a different version of the label distance that depends on the features of the data could be used to remedy this problem, but analyzing the compatibility of such a label distance with the proposed bound requires further research.

### C.1.3 EFFECTIVENESS OF MINIMIZING THE UPPER BOUND IN THEOREM 3 VIA SOLVING EQ. 3

In Fig. 7, we show how the upper bound changes as the optimization progresses for transfer learning from Imagenet to four target tasks with the ResNet-18 model. Similar to experiments in Sec. 4.4 of the paper we optimize over the transformation $A$ while $B$ and $D$ are fixed to a random permutation matrix and the source prior. After about 600 epochs the optimization problem converges to a local minima.

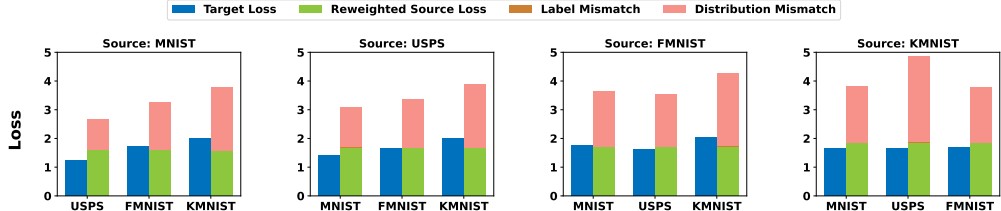

Figure 8: Decomposition of the bound in Theorem 3 into its three components illustrates that the distribution mismatch term explains the difference in transferability. Similar tasks such as USPS and MNIST have the smallest loss after fine-tuning and also have the smallest residual Wasserstein distance, $W_d(S''', T)$, after learning the transformations using our optimization.

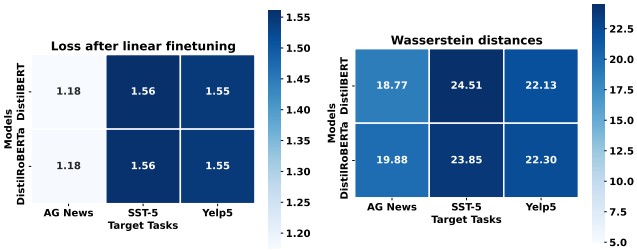

Figure 9: Task relatedness measured by $W_d(P_{S'''}, P_T)$ correlates with transferability (i.e., target loss). A target task related to DBPedia such as AG News achieves a smaller loss and a smaller residual Wasserstein distance compared to less-related tasks such as SST-5 and Yelp5.

## C.2 ADDITIONAL RESULTS FOR THE IMPACT OF TASK RELATEDNESS ON TRANSFERABILITY (SEC. 4.3)

### C.2.1 ADDITIONAL RESULTS FOR IMAGE CLASSIFICATION

Here we provide details of the experiment presented in Sec. 4.3 about the effect of task relatedness on the transferability after linear fine-tuning. Similar to the results presented in Fig. 3 of the main paper the results in Fig. 8 show that when the source and the target tasks are related then both the loss after linear fine-tuning and our bound are small as in the case when the source is MNIST and target is USPS or vice versa. When the target tasks are unrelated to the source data then both the loss after linear fine-tuning and our bound remain the same regardless of the chosen source data. For e.g., when the target task is KMNIST, using MNIST, USPS or FMNIST produces the same loss and the upper bound. Lastly, for any particular source, Fig. 8 shows that the bound only differs because of the differences in the distribution mismatch term which measures $W_d(S''', T)$. This shows that when the distribution mismatch can be minimized, both the empirical and predicted transferability improve, demonstrating transfer between related source and target tasks is both easier and explainable.

### C.2.2 RESULTS FOR NLP SENTENCE CLASSIFICATION TASK

In this section, we use sentence classification NLP task to further demonstrate the effect of task relatedness on transferability and the proposed upper bound. For this experiment, we first fine-tune the entire DistilBERT (Sanh et al., 2020) and DistilRoBERTa (Liu et al., 2019) models distilled on English Wikipedia and Toronto Book Corpus, and OpenAI's WebText dataset, respectively, using a subsample of 10,000 points from the DBPedia dataset. We then use these fine-tuned models to evaluate the transferability to AG news, SST-5, and Yelp datasets. The results in Fig. 9 show that the loss after linear fine-tuning on AG News is the smallest among the three datasets. This coincides with the Wasserstein distance obtained after learning the transformations which explains why transfer to AG News is more successful compared to other datasets. This observation is reasonable, especially considering that both DBPedia and AG News have structured information. Moreover, since DBPedia is related to Wikipedia, the terms and entities appearing in AG News are more related to those

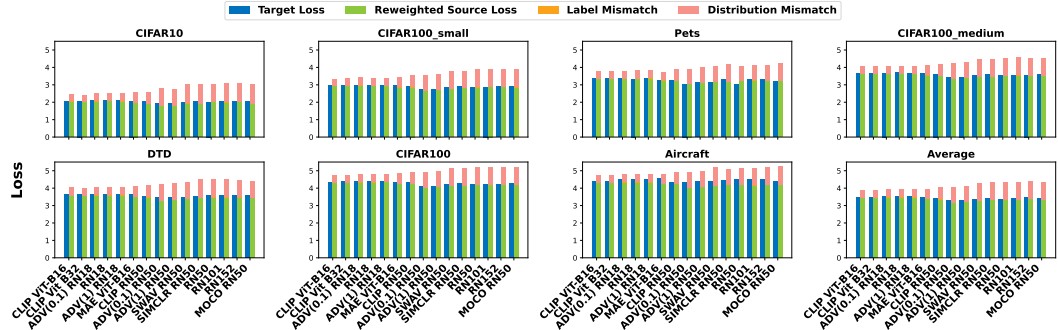

Figure 10: Additional results for comparison of empirical versus predicted transferability for large pre-trained models. Our upper bound proposed in Theorem 3 consistently achieves a small gap to empirical transferability obtained after linear fine-tuning.

appearing in DBPedia in comparison to terms/entities appearing in SST-5 and Yelp which consist of movie reviews and reviews collected from Yelp.

For our experiments, in this section, we follow a similar setting of fixing $B$ to be a random permutation matrix, $C$ to the prior of the source, and only learn the transformation $A$. We sample 10,000 points from DBPedia belonging to the same number of classes as those present in the target task (for e.g., for AG News we sample data from 4 randomly selected classes of DBPedia) and use this data as the source data to train $h_S$ with gradient norm penalty ($\tau = 0.02$). All experiments are run for 3 random seeds and average results are reported in Fig. 9.

### C.3 Additional results for predicting transferability using large pre-trained models (Sec. 4.4)

#### C.3.1 Additional results image classification

Here we present additional results on computer vision classification tasks of predicting transferability through the bound proposed in Theorem 3 which were omitted in the main paper due to space limitation. Consistent with the results shown in Fig. 4 of the main paper, from the results in Fig. 10 we observe that there is a small gap between the loss after linear fine-tuning and the bound for models trained with various pre-training methods and architectures. Moreover, the bound is tighter for models trained with architectures that have a smaller representation space such as ResNet18 and ViT-B-16 which have 512-dimensional representation space. This is attributed to the difficulty of optimizing the Wasserstein distance in higher dimensional representation space.

In Fig. 11, we demonstrate that the proposed upper bound is strongly (negatively) correlated with accuracy after linear fine-tuning (when $h_T$ is trained without the gradient norm penalty). The proposed upper bound correlates negatively since the upper bound predicts the loss (lower is better) but it is compared to accuracy (higher is better). The Pearson correlation coefficient varies from ($-0.69$ to $-0.84$) for the models used in Fig. 4.

#### C.3.2 Results for NLP sentence classification task

In addition to vision tasks, we also evaluate the effectiveness of our optimization at minimizing the upper bound for NLP classification tasks. Similar to Sec. C.2.2, we focus on the task of sentence classification here. For this experiment, we fine-tune the entire DistilBERT (Sanh et al., 2020) and DistilRoBERTa (Liu et al., 2019) models distilled on English Wikipedia and Toronto Book Corpus, and OpenAI's WebText dataset, respectively, using a subsample of 10,000 points from the DBPedia dataset. Using the encoder of this new model, we linearly fine-tune using the data from the target tasks as well as use our optimization to learn the transformations to minimize the upper bound. The results present in Fig. 12, show that our bound consistently achieves a small gap to the loss after linear fine-tuning. These results show that our task transfer analysis can effectively explain transferability across a wide range of tasks.

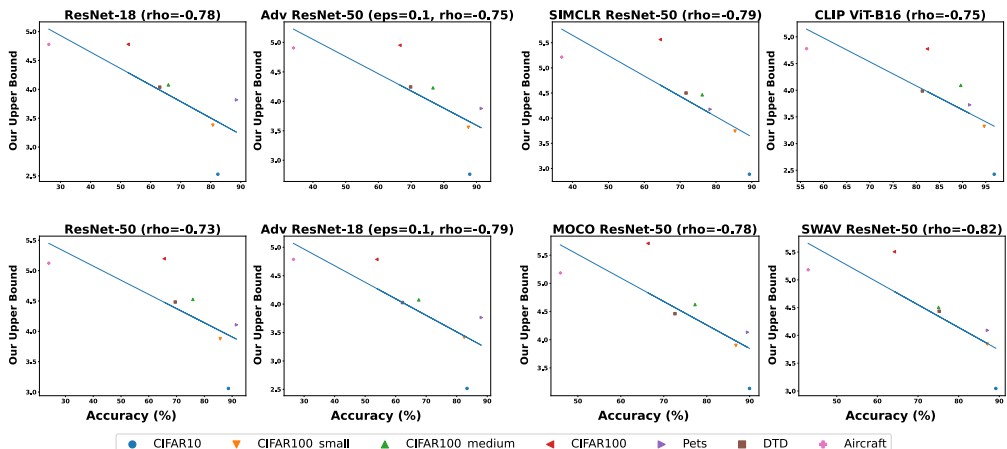

Figure 11: The proposed upper bound on transferability is highly (negatively) correlated with the accuracy of the models after linear fine-tuning. This shows that a task with a smaller upper bound is more transferable. Each subplot shows transfer learning with Imagenet as the source task to various target tasks for a specific model architecture and training method. The Pearson correlation coefficient is reported in the title of each subplot.

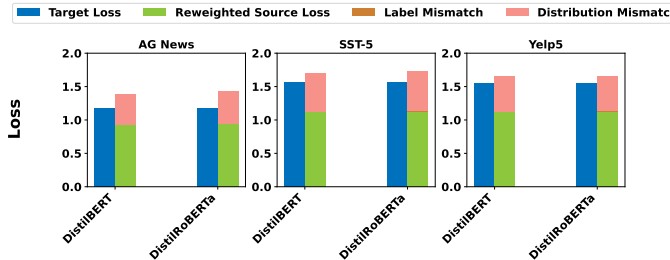

Figure 12: Transferability predicted by our analysis consistently achieves a small gap to empirical transferability even for NLP classification tasks (AG News, SST-5, Yelp5) using DistilBERT and DistilRoBERTa as pre-trained models which are fully fine-tuned with DBPedia.

### C.4 COMPARISON WITH SBTE APPROACHES ON PRETRAINED MODEL SELECTION PROBLEM

In Table 1, we show that our proposed upper bound achieves a high correlation with the accuracy after linear fine-tuning (when $h_T$ is trained without the gradient norm penalty) on pretrained model selection problem using six pretrained models (supervised ResNet-50/101/152, SIMCLR/MOCO/SWAV trained ResNet50). However, since we need to solve Eq. 3 to learn the transformations, our approach is not as computationally efficient as SbTE approaches. We use the official code from You et al. (2021) to compute the scores for NCE, Leep, and LogMe along with the official code of (Shao et al., 2022) for SFDA. For PACTran Ding et al. (2022), we also use their official code with the PACTran-Gaussian method with $N/K = 100, \beta = 10N, \sigma^2 = D/100$ where $N$ denotes the number of samples and $K$ denotes the number of classes. This setting is similar to the PACTran-Gauss$_{fix}$ setting used in their work with the difference that we use $N/K = 100$ so as to use a sufficiently large number of samples, especially considering that all our other results for SbTE methods are computed on the full training set.

### C.5 LIPSCHITZ CONSTRAINED LINEAR FINE-TUNING

#### C.5.1 IMPLEMENTING SOFTMAX CLASSIFICATION WITH $\tau-$LIPSCHITZ LOSS

To use the bound Theorem. 3, it is required that the loss be $\tau-$Lipschitz continuous w.r.t. $z$ in the input domain $\mathcal{Z}$. To enforce this, while learning the weights of the softmax classifier (aka linear

Table 1: The proposed upper bound achieves competitive performance on the pretrained model selection task considered by popular score-based transferability estimation works. Pearson correlation between accuracy after linear fine-tuning and scores from different SbTE methods are reported. For NCE (Tran et al., 2019), Leep (Nguyen et al., 2020), LogMe (You et al., 2021), and SFDA (Shao et al., 2022) a higher correlation is better whereas for PACTran (Ding et al., 2022) and our bound a higher negative correlation is better.

| Target task | NCE | Leep | LogMe | SFDA | PACTran | Ours |
|---|---|---|---|---|---|---|
| CIFAR-100 | 0.98 | 0.67 | 0.87 | 0.93 | -0.86 | -0.66 |
| Pets | 0.94 | -0.24 | 0.87 | -0.19 | -0.54 | -0.64 |
| DTD | 0.09 | 0.81 | 0.76 | 0.05 | 0.63 | -0.72 |

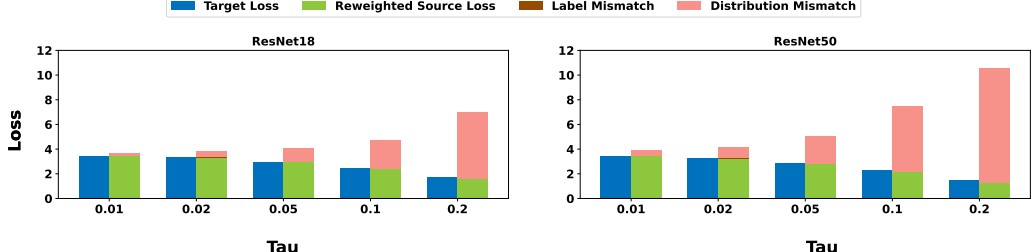

Figure 13: Trade-off between the cross-entropy loss after linear fine-tuning and the upper bound in Theorem 3 as a function of $\tau$, for ResNet18 and ResNet50 models pretrained on Imagenet and linearly fine-tuned on the Pets dataset. Increasing the value of $\tau$ leads to a decrease in the cross-entropy loss after fine-tuning but increases in the proposed bound mainly due to the $\tau \cdot W_d$ term.

fine-tuning) for the source or the target, we add the gradient norm penalty as used in previous works (Shen et al., 2018; Arjovsky et al., 2017) and solve the following optimization problem

$$\min_h \frac{1}{N} \sum_i \left[ \ell(h(z_i), y_i) + \rho \max_y \max\{0, \|\nabla_z \ell(h(z_i), y)\|_2 - \tau\}^2 \right] \quad (\rho \approx 10^4)$$

where $\ell(h(z), y) = -w_y^T z + \log \sum_j e^{w_j^T z}$ is the cross-entropy loss.

### C.5.2 TRADE-OFF BETWEEN EMPIRICAL AND PREDICTED TRANSFERABILITY

Constraining the Lipschitz coefficient of the classifier increases both the target and the source cross-entropy loss since the hypothesis set is being restricted. The smaller the $\tau$ is, the larger the loss becomes. On the other hand, the smaller $\tau$ makes the distribution mismatch term in Theorem 3 also smaller. Since the bound is the sum of the source loss and the distribution mismatch (and label mismatch), there is a trade-off determined by the value of $\tau$. We illustrate the effect of the values of $\tau$ on the empirical and predicted transferability. As mentioned previously, we train both the classifier for the source $h_S$ and the target $h_T$ with an additional penalty on the gradient norm to make them $\tau-$Lipschitz. In Fig. 13, we present results of varying the value of $\tau$ for the transfer to the Pets dataset with Imagenet as the source. For this experiment, we selected 37 random classes from Imagenet and only learned the transform $A$ by keeping $B$ fixed to a random permutation and $C$ fixed to the uniform prior over source classes. We observe that the performance of linear fine-tuning degrades as we decrease the value of $\tau$ but explainability through the bound improves since the distribution mismatch term (dependent on $\tau$) decreases in the bound. However, making $\tau$ too small is not preferable since it leads to an increase in the first term of the bound (re-weighted source loss) increasing the bound overall. Moreover, it also leads to a degradation in the accuracy after linear fine-tuning. For our experiments, we use $\tau = 0.02$ since it doesn't decrease the accuracy of fine-tuning significantly and leads to a small gap between empirical and predicted transferability.

# D    DETAILS OF THE EXPERIMENTS

All codes are written in Python using Tensorflow/Pytorch and were run on an Intel(R) Xeon(R) Platinum 8358 CPU with 200 GB of RAM and an Nvidia A10 GPU. Implementation and hyperparameters are described below. Our codes can be found in the supplementary material.

## D.1    DATASET DETAILS

In our work, we used the standard image classification benchmark datasets along with standard natural language processing datasets[1].

**Aircraft (Maji et al., 2013):** consists of 10,000 aircraft images belonging to 100 classes.

**CIFAR-10/100 (Krizhevsky et al., 2009):** These datasets contain 60,000 images belonging to 10/100 categories. Additionally, we created two subsets of CIFAR100 with the first 25 (small) and 50 (medium) classes.

**DTD(Cimpoi et al., 2014):** consists of 5,640 textural images belonging to 47 categories.

**Fashion MNIST (Xiao et al., 2017):** consists of 70,000 grayscale images belonging to 10 categories.

**Pets (Parkhi et al., 2012):** consists of 7049 images of Cats and Dogs spread across 37 categories.

**Imagenet (Deng et al., 2009):** consists of 1.1 million images belonging to 1000 categories.

**Yelp (Zhang et al., 2015):** consists of 650,000 training and 50,000 test examples belonging to 5 classes.

**Stanford Sentiment Treebank (SST-5) (Zhang et al., 2015):** consists of 8,544 training and 2,210 test samples belonging to 5 classes.

**AG News (Zhang et al., 2015):** consists of 120,000 training and 7,600 test examples belonging to 4 classes

**DBPedia (Zhang et al., 2015):** consists of 560,000 training and 70,000 test examples belonging to 14 classes

## D.2    SEMANTICALLY SIMILAR CLASSES FOR CIFAR-10 AND PETS FROM IMAGENET

For our experiments with CIFAR-10 in Sec. 4.2, we selected the following semantically similar classes from Imagenet, `{airliner, minivan, cock, tabby cat, ox, chihuahua, bullfrog, sorrel, submarine, fire engine}`. For our experiments with the Pets dataset in App. C.1.2, we selected the following classes for Dogs `{boston bull, miniature schnauzer, giant schnauzer, standard schnauzer, scotch terrier, chrysanthemum dog, silky terrier, a soft-coated wheaten terrier, west Highland white terrier, lhasa, lat-coated retriever, curly-coated retriever, golden retriever, labrador retriever, chesapeake bay retriever, german short-haired pointer, vizsla, hungarian pointer, english setter, irish setter, gordon setter, brittany spaniel, clumber, english springer, welsh springer spaniel, cocker spaniel}` and the following for Cats `{tabby, tiger cat, persian cat, siamese cat, egyptian cat, cougar, lynx, leopard, snow leopard, jaguar, lion, tiger}`. Since some species of Cats and Dogs present in the Pets dataset are not present in Imagenet, we select broadly related classes for our experiments.

## D.3    ADDITIONAL EXPERIMENTAL DETAILS

In our experiments, in Sec. 4.4, we used pre-trained models available from Pytorch for ResNet18/50, along with publically available pre-trained models provided in the official repositories of each training method. For each experiment, we subsample data from the Imagenet dataset belonging to the same

---

[1]All NLP datasets and models are obtained from `https://huggingface.co/`.

number of classes as those present in the target dataset and use this data to train the linear layer on top of the representations extracted from the pre-trained model along with a gradient norm penalty. To speed up the experiments, we use only 10,000 points from the subsample of Imagenet for training the linear classifier and computing the transfer. For evaluation, we use a similar subsample of the validation dataset of Imagenet containing all the samples belonging to the subsampled classes. Fine-tuning on this dataset takes about 0.05 seconds per epoch for the task of transfer from Imagenet to Pets with the ResNet-18 model (we run the fine-tuning for a total of 5000 epochs).

Along with training the linear classifiers with a gradient norm penalty, we standardize the features extracted from the pre-trained models to remove their mean (along each axis) and make them have a unit standard deviation (along each axis). While standardizing the features do not have a significant impact on the loss of the classifiers, including it makes it easier to match the distributions of the source and target data after transformations. Since our optimization problem transforms the source distribution to match the distribution of the target by solving the optimization problem in Eq. 3 by working on mini-batches, it is important that the size of the batch be greater than the dimension of the representation space of the pre-trained encoder. For e.g., for ResNet18 models which have a representation dimension of 512, we use a batch size of 1000 and for ResNet50 models which have a representation dimension of 2048, we use a batch size of 2500. Having a smaller batch size than the dimension could lead to a noisy gradient since for that batch the transformation can achieve a perfect matching, which may not generalize to data from other batches or unseen test data.

While computing the transformations, we apply the same augmentation (re-sizing and center cropping)/normalization to the training data as those applied to the test data. Along with this, we extract the features of the training and test data from the pre-trained model once and use these to train the linear layer. We note that this is done to save the computation time and better results could be obtained by allowing for extracting features after data augmentation for every batch.

Finally, for our experiments in Sec. 4.3, the encoders are trained end-to-end on the source task. This is in contrast to our other experiments where the encoders are pre-trained and data from the source task is only used for linear fine-tuning. This is done since there are no pre-trained models available for MNIST-type tasks considered in this section and training a model on these datasets is relatively easy and cheap. Using these models, task relatedness is evaluated by fine-tuning a linear layer using the data from the target task as well as the transformations are computed by solving Eq. 3. We used $\tau = 0.2$ here. We run the experiments with 3 random seeds and report the average results.

