# OpenReview forum: "Analysis of Task Transferability in Large Pre-trained Classifiers"
_ICLR.cc/2024/Conference — ICLR 2024 Conference Withdrawn Submission_

### Official Review · Reviewer_nX5m · 2023-10-28

**Soundness:** 3 good
**Presentation:** 3 good
**Contribution:** 2 fair
**Rating:** 5
**Confidence:** 4

**Summary:**

This paper provides a theoretical framework for understanding transfer learning. The authors decompose the transfer learning loss into three parts, namely the losses incurred due to the prior transform, feature transform, and distribution mismatch. Building on top of this decomposition, the authors provide a loss upper bound on the transfer learning, and empirically demonstrate that this bound is close to what happens in the real world.

**Strengths:**

- The idea of decomposing the transfer learning loss into the three parts is novel and interesting.
- The paper is well-written with the motivation for each part of the decomposition clearly stated.
-  The empirical verification of the bound is nice.

**Weaknesses:**

- The current framework doesn’t incorporate the effect of network architecture, which seems to be quite important in the real world.
- It’s unclear what’s something that is constructive out of this analysis. The authors don’t provide any suggestion on how to improve the transfer learning performance based on the current analysis framework.
- It would be good to provide some outline of the key steps of the theoretical derivation. Given that this is mainly a theory paper, it would be good to highlight the most non-trivial / novel part of the analysis.

**Questions:**

- How do you think of the relative importance of these different types of mismatch? Is there any way to directly verify them in the real world?

---

### Official Review · Reviewer_Zr88 · 2023-10-28

**Soundness:** 2 fair
**Presentation:** 2 fair
**Contribution:** 2 fair
**Rating:** 3
**Confidence:** 4

**Summary:**

This paper delves into the domain of transfer learning, specifically the relationship between performance on a source task and its consequent performance on a target task after linear fine-tuning. The authors aim to understand the transferability of large pre-trained models when applied to downstream classification tasks. Their claimed contributions include:

1. Introduction of the **Task Transfer Analysis** approach, which alters the distribution and classifier of the source task to mirror those of the target task.
2. Proposal of an **upper bound on transferability**. This is a composite measure derived from:
   - The conditional entropy between the label distributions of the two tasks.
   - The weighted loss of the source classifier on the source task.
   - The Wasserstein distance between the modified source and the target distributions.
3. Formulation of an **optimization problem** designed to minimize the aforementioned bound to determine transferability.

The empirical results, obtained from top-tier pre-trained models, suggest that this proposed upper bound can provide a reasonable estimate of transferability across various datasets. A key takeaway from the study is : the more seamlessly the source task can be modified to match the target, the better the transferability.

Future research, as suggested by the authors, may involve extending this analysis to encompass full fine-tuning and exploring alternate loss functions.

**Strengths:**

1. The work introduces a new "Task Transfer Analysis" approach that aims to transform both the distribution and the classifier of the source task to resemble those of the target task.

2. The paper proposes an upper bound on transferability, allowing for a more quantifiable understanding of how one task relates to another in the context of transfer learning. This upper bound considers the Wasserstein distance, conditional entropy, and the weighted loss, offering a multi-faceted perspective on transferability.

3. The authors further introduce an optimization problem designed to minimize the proposed bound, providing a practical tool or method for estimating transferability.

4. The paper covers many datasets and pre-trained models. This broad coverage enhances the generalizability of the findings and ensures they're applicable across diverse contexts.

**Weaknesses:**

1. **Scope Limitation**: A glaring limitation of this study lies in its exclusive focus on linear fine-tuning. The practical implications of linear fine-tuning in transfer learning are limited. If the overarching goal is to gauge performance, why not simply engage in the linear fine-tuning itself? When distilled to its essence, with features in hand, linear fine-tuning is fundamentally a softmax regression challenge—a territory that has been well-studied in literature with efficient second-order solutions at its disposal. The potential value of this paper could have been augmented significantly had it considered a more comprehensive exploration of general fine-tuning, or what the authors label as "full fine-tuning".

2. **Insufficient Comparative Analysis**: The paper exhibits a noticeable deficiency in terms of benchmarking against established literature. While there's frequent phrases against score-based transferability estimation, the main content conspicuously lacks any substantive comparative evaluations. The only claim that their approach offers insights unattainable through SbTE methods feels unsubstantiated, especially considering existing studies such as "Representation similarity analysis for efficient task taxonomy & transfer learning" and "Duality Diagram Similarity: A Generic Framework for Initialization Selection in Task Transfer Learning". These studies have delved deep into elucidating transferability using both source and target tasks, casting doubt on the novelty of the authors' task transfer analysis. (The authors cited these two papers, but failed to understand the methods.)

3. **Inefficiency and Instability Concerns**: Turning our attention to Figure 2, it is evident that the chosen subset of source data can have significant influence over the upper bound. Given that state-of-the-art pre-trained models often draw upon vast datasets of pre-training images, often running into millions, the method proposed by the authors appears neither resource-efficient nor consistently stable. Such a scalability challenge hinders the broader applicability of their approach in real-world scenarios where data volume is massive.

In summary, while the paper takes an interesting approach to task transferability analysis, it seems to be weak by its limited scope, a lack of rigorous comparative assessment, and potential challenges in scalability and stability. Expanding the study's scope to encompass broader fine-tuning techniques and integrating more robust comparative metrics would substantially improve its academic and practical significance.

**Questions:**

1. **Regarding Figures 2 and 4:**
   - In both Figure 2 and Figure 4, the legends indicate the presence of three components: "Reweighted Source Loss", "Label Mismatch", and "Distribution Mismatch". However, upon inspection, there seems to be no visible representation for "Label Mismatch", i.e. the orange segment. Can the authors clarify this discrepancy?

2. **Concerning Figure 4:**
   - It is noticeable that nearly all the models in Figure 4 exhibit almost identical target losses. This is counterintuitive given the presumption that models with varying architectures and training methods should manifest disparate downstream transfer performances. Could the authors provide an explanation or rationale behind this observation?

3. **On the Paper's Textual Content:**
   - There appears to be a repetitive phrasing in the sentence, "Our approach works by transforming the source distribution (and the classifier of the source task) **by transforming**." The consecutive use of "by transforming" seems redundant. Is it a typo?

4. **Regarding Terminology:**
   - The chosen abbreviation for "score-based transferability estimation" seems unconventional. Could the authors justify this choice or consider a more intuitive abbreviation?

While the paper presents valuable insights on task transferability, it would benefit from addressing the above queries to ensure clarity and coherence for readers.

**Details Of Ethics Concerns:**

Not applicable.

---

### Official Review · Reviewer_2KtK · 2023-10-30

**Soundness:** 2 fair
**Presentation:** 2 fair
**Contribution:** 2 fair
**Rating:** 3
**Confidence:** 4

**Summary:**

This paper proposes a method to estimate the transferability of a pre-trained classifier for a given task without directly conducting transfer learning. Specifically, it derives an upper bound on the observed loss of the model after linear fine-tuning by solving the proposed optimization problem. This optimization problem includes prior transformation, label transformation, and feature transformation, incorporating a term that accounts for the mismatch between the data distribution after applying their combined transformations and the target distribution. The authors demonstrate the proximity of the proposed upper bound to the target loss through simple experiments.

**Strengths:**

1. This paper is overall clearly clarified and well organized.
2. This paper provides both theoretical and empirical analyses.

**Weaknesses:**

1. The practicality of the proposed method is limited:
 - Only linear fine-tuning is considered.
 - For the proposed method, not only pre-trained models but also supervised pre-training data are required.
 - When the pre-training dataset is not clearly defined in terms of the classification task (such as the vision-language pre-training dataset), the application of the proposed method is not straightforward.
2. The proposed method is not robust. In other words, depending on the selection of source classes, significantly different estimations of transferability can be made for the same pre-trained model. For instance, as demonstrated in Section 4.2, considering a higher number of source classes than downstream classes leads to an increase in the estimated loss upper bound. In addition, sampling challenging classes among the source classes leads to a higher estimated upper bound.

**Questions:**

See Weaknesses

---

### Official Review · Reviewer_GYU6 · 2023-11-05

**Soundness:** 2 fair
**Presentation:** 2 fair
**Contribution:** 2 fair
**Rating:** 3
**Confidence:** 4

**Summary:**

This paper analyzes the transferability of pre-trained models on downstream classification tasks after linear fine-tuning. It transforms the distribution (and classifier) of the source task to produce a new distribution (and classifier) similar to that of the target task. Based on this, it proposes an upper bound on transferability composed of the Wasserstein distance between the transformed source and the target distributions, the conditional entropy between the label distributions of the two tasks, and the weighted loss of the source classifier on the source task. It then proposes an optimization problem that minimizes the proposed bound to estimate transferability. Experiments on some image or sentence classification tasks with different pre-trained models show that the proposed upper bound can estimate the performance on the target tasks after linear fine-tuning.

**Strengths:**

S1: This idea of considering relatedness between the source and target tasks for transferability makes sense. The general framework designs of prior transformation, label transformation, feature transformation and distribution matching seem reasonable.

S2: The method is evaluated on various pre-trained models on both image and sentence classification tasks.

**Weaknesses:**

W1: The aim of the paper seems to estimate the bound of target performance after learning a linear classifier with a fixed pre-trained network, and the estimation needs source data, target data and the pre-trained model. I have some concerns about the problem setting. (1) Why don’t we directly train a linear classifier with the target data to get the performance (since we already have target data and training a linear classifier does not have a high cost)? What is the point of estimating the bound? (2) The requirement of source data prevents practical usage of this setting, since many pre-trained models do not provide their original training data with them. (3) This paper focuses on a fixed pre-trained model, which still has a gap between more common practices such as fine-tuning or parameter-efficient fine-tuning. (4) The problem here can only be used in classification tasks.

W2: The general estimation framework makes sense to me, but the technical designs of each step are not novel or insightful enough to me. The designs of prior transform, label transform, and feature transform are straightforward, and the Wasserstein distance has already been well studied in related topics such as distribution shifts and domain adaptation.

W3: The experimental part is hard to follow in some ways. (1) Why should we select the classes from the source label set for estimation instead of using the whole label set? If the label or semantic shift matters for transferability, why do random selections and semantic selections have similar results? (2) The method contains four steps of prior transformation, label transformation, feature transformation, and distribution matching, but there are no direct ablation studies to evaluate these four parts. (3) In Section 4.4, ImageNet is used as the source task, but some of the pre-trained models, such as CLIP, are not pre-trained with ImageNet. (4) In Figures 2 and 4, there are no bars corresponding to ‘Label Mismatch’. (5) It is claimed that the estimation of the proposed method has a small gap with the target loss. However, there are no other reference results to compare with, so how can we define ‘a small gap’?

**Questions:**

Could the method in this paper provide more accurate model selection results compared with existing score-based transferability estimation (SbTE) methods?